# Perceptual Fairness in Image Restoration

**Guy Ohayon**
Faculty of Computer Science
Technion–Israel Institute of Technology
`ohayonguy@cs.technion.ac.il`

**Michael Elad**
Faculty of Computer Science
Technion–Israel Institute of Technology
`elad@cs.technion.ac.il`

**Tomer Michaeli**
Faculty of Electrical and Computer Engineering
Technion–Israel Institute of Technology
`tomer.m@ee.technion.ac.il`

## Abstract

Fairness in image restoration tasks is the desire to treat different sub-groups of images equally well. Existing definitions of fairness in image restoration are highly restrictive. They consider a reconstruction to be a correct outcome for a group (*e.g.*, women) *only* if it falls within the group's set of ground truth images (*e.g.*, natural images of women); otherwise, it is considered *entirely* incorrect. Consequently, such definitions are prone to controversy, as errors in image restoration can manifest in various ways. In this work we offer an alternative approach towards fairness in image restoration, by considering the *Group Perceptual Index* (GPI), which we define as the statistical distance between the distribution of the group's ground truth images and the distribution of their reconstructions. We assess the fairness of an algorithm by comparing the GPI of different groups, and say that it achieves perfect *Perceptual Fairness* (PF) if the GPIs of all groups are identical. We motivate and theoretically study our new notion of fairness, draw its connection to previous ones, and demonstrate its utility on state-of-the-art face image restoration algorithms.

## 1 Introduction

Tremendous efforts have been dedicated to understanding, formalizing, and mitigating fairness issues in various tasks, including classification [17, 22, 29, 81, 94, 95], regression [2, 7, 8, 12, 43, 65], clustering [4–6, 13, 70, 73], recommendation [25, 26, 46, 52, 92], and generative modeling [15, 24, 44, 66, 74, 75, 96]. Fairness definitions remain largely controversial, yet broadly speaking, they typically advocate for independence (or conditional independence) between sensitive attributes (ethnicity, gender, *etc*.) and the predictions of an algorithm. In classification tasks, for instance, the input data carries sensitive attributes, which are often required to be statistically independent of the predictions (*e.g.*, deciding whether to grant a loan should not be influenced by the applicant's gender). Similarly, in text-to-image generation, fairness often advocates for statistical independence between the sensitive attributes of the generated images and the text instruction used [24]. For instance, the prompt ``An image of a firefighter'' should result in images featuring people of various genders, ethnicities, *etc*.

While fairness is commonly associated with the desire to *eliminate* the dependencies between sensitive attributes and the predictions, fairness in image restoration tasks (*e.g.*, denoising, super-resolution) has a fundamentally different meaning. In image restoration, *both* the input and the output carry sensitive attributes, and the goal is to *preserve* the attributes of different groups equally well [34]. But what exactly constitutes such a preservation of sensitive attributes? Let us denote by $x$, $y$, and $\hat{x}$ the unobserved source image, its degraded version (*e.g.*, noisy, blurry), and the reconstruction of $x$

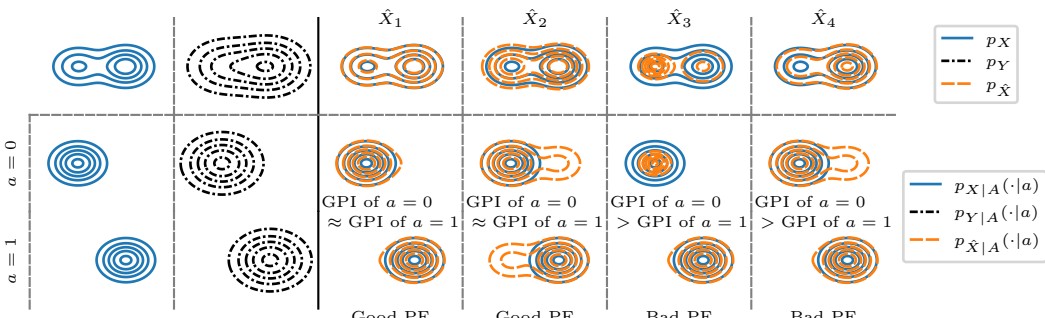

**Figure 1:** Illustrative example of the proposed notion of Perceptual Fairness (PF). This figure presents four possible restoration algorithms exhibiting different behaviors and fairness performance. In this example, the sensitive attribute $A$ takes the values 0 or 1 with probabilities $P(A = 0) < P(A = 1)$. The distributions $p_X$ and $p_Y$ correspond to the ground truth signals (*e.g.*, natural images) and their degraded measurements (*e.g.*, noisy images), respectively. The distribution $p_{X|A}(\cdot|a)$ corresponds to the ground truth signals associated with the attribute value $a$, and $p_{Y|A}(\cdot|a)$ is the distribution of their degraded measurements. The distribution of all reconstructions is denoted by $p_{\hat{X}}$, and $p_{\hat{X}|A}(\cdot|a)$ is the distribution of the reconstructions associated with attribute value $a$. The Group Perceptual Index (GPI) of the group associated with $a$ is defined as the statistical distance between $p_{\hat{X}|A}(\cdot|a)$ and $p_{X|A}(\cdot|a)$, and good PF is achieved when the GPIs of all groups are (roughly) similar. For example, $\hat{X}_1$ achieves good PF since the GPIs of both $a = 0$ and $a = 1$ are roughly equal, while $\hat{X}_3$ achieves poor PF since the GPI of $a = 0$ is worse (larger) than that of $a = 1$. See Section 2 for more details.

from $y$, respectively. Additionally, let $\mathcal{X}_a$ denote the set of images $x$ carrying the sensitive attributes $a$. Jalal et al. [34] deem the reconstruction of any $x \in \mathcal{X}_a$ as correct only if $\hat{x} \in \mathcal{X}_a$. This allows practitioners to evaluate fairness in an intuitive way, by classifying the reconstructed images produced for different groups. For instance, regarding $x$, $y$, and $\hat{x}$ as realizations of random vectors $X$, $Y$, and $\hat{X}$, respectively, Representation Demographic Parity (RDP) states that $\mathbb{P}(\hat{X} \in \mathcal{X}_a | X \in \mathcal{X}_a)$ should be the same for all $a$, and Proportional Representation (PR) states that $\mathbb{P}(\hat{X} \in \mathcal{X}_a) = \mathbb{P}(X \in \mathcal{X}_a)$ should hold for every $a$. However, the idea that a reconstructed image $\hat{x}$ can either be an *entirely correct* output ($\hat{x} \in \mathcal{X}_a$) or an *entirely incorrect* output ($\hat{x} \notin \mathcal{X}_a$) is highly limiting, as errors in image restoration can manifest in many different ways. Indeed, what if one algorithm always produces blank images given inputs from a specific group, and another algorithm produces images that are "almost" in $\mathcal{X}_a$ for such inputs (*e.g.*, each output is only close to some image in $\mathcal{X}_a$)? Should both algorithms be considered equally (and completely) erroneous for that group? Furthermore, quantities of the form $\mathbb{P}(\hat{X} \in \mathcal{X}_a | \cdot)$ completely neglect the *distribution* of the images within $\mathcal{X}_a$. For example, assuming the groups are women and non-women, an algorithm that always outputs the same image of a woman when the source image is a woman, but produces diverse non-women images when the source is not a woman, still satisfies RDP. Does this algorithm truly treat women fairly?

To address these controversies, we propose to examine how the restoration method affects the *distribution* of each group of interest (*e.g.*, the distribution of images of women or non-women). Specifically, we define the *Group Perceptual Index* (GPI) to be the statistical distance (*e.g.*, Wasserstein) between the distribution of the group's ground truth images and the distribution of their reconstructions. We then associate *Perceptual Fairness* (PF) with the degree to which the GPIs of the different groups are close to one another. In other words, the PF of an algorithm corresponds to the parity among the GPIs of the groups of interest (see Figure 1 for intuition). The rationale behind using such an index is two-fold. First, it solves the aforementioned controversies. For example, an algorithm that always outputs the same image of a woman when the source image is a woman, and diverse non-women images otherwise, would achieve poor GPI for women and good GPI for non-women, thus resulting in poor PF. Second, the GPI reflects the ability of humans to distinguish between samples of a group's ground truth images and samples of the reconstructions obtained from the degraded images of that group [10]. Thus, achieving good PF (*i.e.*, parity in the GPIs) suggests that this ability is the same for all groups.

This paper is structured as follows. In Section 2 we formulate the image restoration task and present the mathematical notations necessary for this paper. This includes a review of prior fairness definitions in image restoration, alongside our proposed definition. We also discuss why PF can be considered as a generalization of RDP. In Section 3 we present our theoretical findings. For instance, we prove that

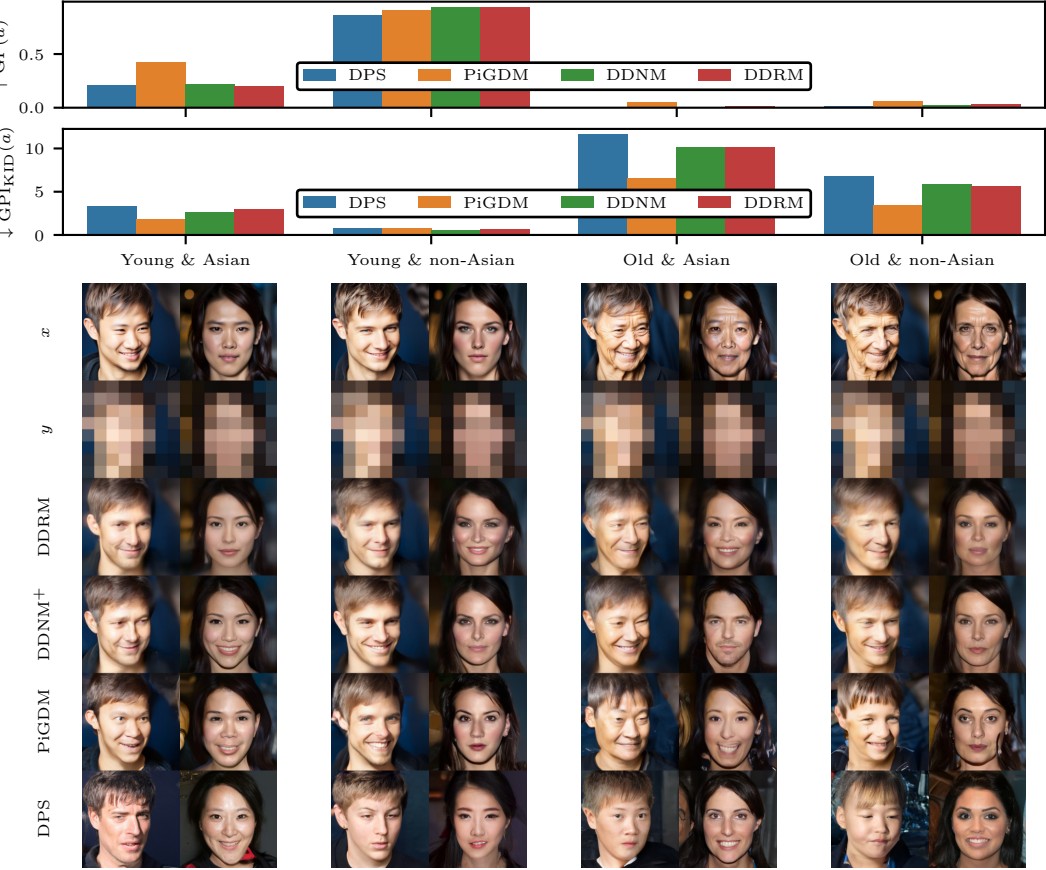

**Figure 2:** Examining fairness in face image super-resolution techniques through the lens of RDP [34] or PF (our proposed notion of fairness). Both RDP and PF assess how well an algorithm treats different fairness groups. Specifically, RDP evaluates the parity in the GP of different groups (higher GP is better), and PF evaluates the parity in the GPI of different groups (lower GPI is better). The results show that the groups old&Asian and old&non-Asian attain similar treatment according to RDP (similar GP scores that are roughly zero), while the latter group attains better treatment according to PF. In Section 4 and Appendix G.7, we show why this outcome of PF is the desired one.

achieving perfect GPI for all groups simultaneously is not feasible when the degradation is sufficiently severe. We also establish an interesting (and perhaps counter-intuitive) relationship between the GPI of different groups for algorithms attaining a perfect Perceptual Index (PI) [10], and show that PF and the PI are often at odds with each other. In Section 4 we demonstrate the practical advantages of PF over RDP. In particular, we show that PF detects bias in cases where RDP fails to do so. Lastly, in Section 5 we discuss the limitations of this work and propose ideas for the future.

## 2 Problem formulation and preliminaries

We adopt the Bayesian perspective of inverse problems, where an image $x$ is regarded as a realization of a random vector $X$ with probability density function $p_X$. Consequently, an input $y$ is a realization of a random vector $Y$ (*e.g.*, a noisy version of $X$), which is related to $X$ via the conditional probability density function $p_{Y|X}$. The task of an estimator $\hat{X}$ (in this paper, an image restoration algorithm) is to estimate $X$ *only* from $Y$, such that $X \to Y \to \hat{X}$ is a Markov chain ($X$ and $\hat{X}$ are statistically independent given $Y$). Given an input $y$, the estimator $\hat{X}$ generates outputs according to the conditional density $p_{\hat{X}|Y}(\cdot|y)$.

## 2.1 Perceptual index

A common way to evaluate the quality of images produced by an image restoration algorithm is to assess the ability of humans to distinguish between samples of ground truth images and samples of the algorithm's outputs. This is typically done by conducting experiments where human observers vote on whether the generated images are real or fake [18, 20, 28, 32, 33, 72, 101, 102]. Importantly, this ability can be quantified by the *Perceptual Index* [10], which is the statistical distance between the distribution of the source images and the distribution of the reconstructed ones,

$$\text{PI}_d := d(p_X, p_{\hat{X}}),\tag{1}$$

where $d(\cdot, \cdot)$ is some divergence between distributions (Kullback–Leibler divergence, total variation distance, Wassersterin distance, *etc.*).

## 2.2 Fairness

### 2.2.1 Previous notions of fairness

Jalal et al. [34] introduced three pioneering notions of fairness for image restoration algorithms: Representation Demographic Parity (RDP), Proportional Representation (PR), and Conditional Proportional Representation (CPR). Formally, given a collection of sets of images $\{\mathcal{X}_{a_i}\}_{i=1}^k$, where $a_i$ is a vector of sensitive attributes and each $\mathcal{X}_{a_i}$ represents the group carrying the sensitive attributes $a_i$, these notions are defined by

$$\text{RDP: } \mathbb{P}(\hat{X} \in \mathcal{X}_{a_i} | X \in \mathcal{X}_{a_i}) = \mathbb{P}(\hat{X} \in \mathcal{X}_{a_j} | X \in \mathcal{X}_{a_j}) \text{ for every } i, j;\tag{2}$$

$$\text{PR: } \mathbb{P}(\hat{X} \in \mathcal{X}_{a_i}) = \mathbb{P}(X \in \mathcal{X}_{a_i}) \text{ for every } i;\tag{3}$$

$$\text{CPR: } \mathbb{P}(\hat{X} \in \mathcal{X}_{a_i} | Y = y) = \mathbb{P}(X \in \mathcal{X}_{a_i} | Y = y) \text{ for every } i, y.\tag{4}$$

While such definitions are intuitive and practically appealing, they have several limitations. First, any reconstruction that falls even "slightly off" the set $\mathcal{X}_{a_i}$ is considered an entirely wrong outcome for its corresponding group. In other words, reconstructions with minor errors are treated the same as completely wrong ones. Second, these definitions neglect the *distribution* of the groups' images. Consequently, an algorithm can satisfy RDP, PR, CPR, *etc.*, while treating some groups much worse than others in terms of the *statistics* of the reconstructed images. For instance, consider dogs and cats as the two fairness groups. Let $\mathcal{X}_{\text{dogs}}$ and $\mathcal{X}_{\text{cats}}$ be the sets of images of dogs and cats, respectively, and let $x_{\text{dog}} \in \mathcal{X}_{\text{dogs}}$ be a particular image of a dog. Furthermore, suppose that the species can be perfectly identified from any degraded measurement, *i.e.*,

$$\mathbb{P}(X \in \mathcal{X}_{\text{dogs}} | Y = y) = 1 \text{ or } \mathbb{P}(X \in \mathcal{X}_{\text{cats}} | Y = y) = 1\tag{5}$$

for every $y$. Now, suppose that $\hat{X}$ always produces the image $x_{\text{dog}}$ from any degraded dog image, while generating diverse, high-quality cat images from any degraded cat image. Namely, for every $y$, we have

$$1 = \mathbb{P}(\hat{X} = x_{\text{dog}} | X \in \mathcal{X}_{\text{dogs}}) = \mathbb{P}(\hat{X} \in \mathcal{X}_{\text{dogs}} | X \in \mathcal{X}_{\text{dogs}}) = \mathbb{P}(\hat{X} \in \mathcal{X}_{\text{cats}} | X \in \mathcal{X}_{\text{cats}}),\tag{6}$$

$$\mathbb{P}(\hat{X} = x_{\text{dog}} | Y = y) = \mathbb{P}(\hat{X} \in \mathcal{X}_{\text{dogs}} | Y = y) = \mathbb{P}(X = \mathcal{X}_{\text{dogs}} | Y = y),\tag{7}$$

$$\mathbb{P}(\hat{X} \in \mathcal{X}_{\text{cats}} | Y = y) = \mathbb{P}(X = \mathcal{X}_{\text{cats}} | Y = y).\tag{8}$$

Although this algorithm satisfies RDP (Equation (6)) and CPR (Equations (7) and (8)), which entails PR [34], it is clearly useless for dogs. Should such an algorithm really be deemed as fair, then?

To address such controversies, we propose to represent each group by the *distribution* of their images, and measure the representation error of a group by the extent to which an algorithm "preserves" such a distribution. This requires a more general formulation of fairness groups, which is provided next.

### 2.2.2 Rethinking fairness groups

We denote by $A$ (a random vector) the sensitive attributes of the degraded measurement $Y$, so that $p_{Y|A}(\cdot | a)$ is the distribution of degraded images associated with the attributes $A = a$ (*e.g.*, the distribution of noisy women images). Consequently, the distribution of the ground truth images that possess the sensitive attributes $a$ is given by $p_{X|A}(\cdot | a)$, and the distribution of their reconstructions is

given by $p_{\hat{X}|A}(\cdot|a)$. Moreover, we assume that $A \to Y \to \hat{X}$ forms a Markov chain, implying that knowing $A$ does not affect the reconstructions when $Y$ is given. This assumption is not limiting, since image restoration algorithms are mostly designed to estimate $X$ solely from $Y$, without taking the sensitive attributes as an additional input. See Figure 1 for an illustrative example of the proposed formulation.

Note that such a formulation is quite general, as it does not make any assumptions regarding the nature of the image distributions, whether they have overlapping supports or not, *etc*. Our formulation also generalizes the previous notion of fairness groups, which considers only the support of $p_{X|A}(\cdot|a)$ for every $a$. Indeed, one can think of $\mathcal{X}_a = \mathrm{supp}\, p_{X|A}(\cdot|a)$ as the set of images corresponding to some group, and of $\{\mathcal{X}_a\}_{a \in \mathrm{supp}\, p_A}$ as the collection of all sets. Furthermore, notice that $A$ can also be the degraded measurement itself, *i.e.* $A = Y$. In this case, $p_{X|A}(\cdot|a) = p_{X|Y}(\cdot|a)$ is the posterior distribution of ground truth images given the measurement $a$, and $p_{\hat{X}|A}(\cdot|a) = p_{\hat{X}|Y}(\cdot|a)$ is the distribution of the reconstructions of the measurement $a$. Namely, our mathematical formulation is adaptive to the granularity of fairness groups considered.

### 2.2.3 Perceptual fairness

We define the fairness of an image restoration algorithm as its ability to equally preserve the distribution $p_{X|A}(\cdot|a)$ across all possible values of $a$. Formally, we measure the extent to which an algorithm $\hat{X}$ preserves this distribution by the *Group Perceptual Index*, defined as

$$\mathrm{GPI}_d(a) := d(p_{X|A}(\cdot|a), p_{\hat{X}|A}(\cdot|a)), \tag{9}$$

where $d(\cdot, \cdot)$ is some divergence between distributions. Then, we say that $\hat{X}$ achieves perfect *Perceptual Fairness* with respect to $d$, or perfect $\mathrm{PF}_d$ in short, if

$$\mathrm{GPI}_d(a_1) = \mathrm{GPI}_d(a_2) \tag{10}$$

for every $a_1, a_2 \in \mathrm{supp}\, p_A$ (see Figure 1 to gain intuition). In practice, algorithms may rarely achieve exactly perfect $\mathrm{PF}_d$, while the $\mathrm{GPI}_d$ of different groups may still be roughly equal. In such cases, we say that $\hat{X}$ achieves good $\mathrm{PF}_d$. In contrast, if there exists at least one group that attains far worse $\mathrm{GPI}_d$ than some other group, we say that $\hat{X}$ achieves poor/bad $\mathrm{PF}_d$. Importantly, note that achieving good $\mathrm{PF}_d$ does not necessarily indicate good $\mathrm{PI}_d$ and/or good $\mathrm{GPI}_d$ values.

### 2.2.4 Group Precision, Group Recall, and connection to RDP

In addition to the $\mathrm{PI}_d$ defined in (1), the performance of image restoration algorithms is often measured via the following complementary measures [45, 71]: (1) *Precision*, which is the probability that a sample from $p_{\hat{X}}$ falls within the support of $p_X$, $\mathbb{P}(\hat{X} \in \mathrm{supp}\, p_X)$, and (2) *Recall*, which is the probability that a sample from $p_X$ falls within the support of $p_{\hat{X}}$, $\mathbb{P}(X \in \mathrm{supp}\, p_{\hat{X}})$. Achieving low precision implies that the reconstructed images may not always appear as valid samples from $p_X$. Thus, precision reflects the perceptual *quality* of the reconstructed images. Achieving low recall implies that some portions of the support of $p_X$ may never be generated as outputs by $\hat{X}$. Hence, recall reflects the perceptual *variation* of the reconstructed images.

Since here we are interested in the perceptual quality and the perceptual variation of a *group's* reconstructions, let us define the *Group Precision* and the *Group Recall* by

$$\mathrm{GP}(a) := \mathbb{P}(\hat{X} \in \mathcal{X}_a | A = a), \tag{11}$$

$$\mathrm{GR}(a) := \mathbb{P}(X \in \hat{\mathcal{X}}_a | A = a), \tag{12}$$

where $\mathcal{X}_a = \mathrm{supp}\, p_{X|A}(\cdot|a)$ and $\hat{\mathcal{X}}_a = \mathrm{supp}\, p_{\hat{X}|A}(\cdot|a)$. Hence, when adopting our formulation of fairness groups, satisfying RDP simply means that the GP values of all groups are the same. However, as hinted in previous sections, two groups with similar GP values may still differ significantly in their GR. From the following theorem, we conclude that attaining perfect $\mathrm{PF}_{d_{\mathrm{TV}}}$, where $d_{\mathrm{TV}}(p, q) = \frac{1}{2} \int |p(x) - q(x)| dx$ is the total variation distance between distributions, guarantees that *both* the GP and the GR of all groups have a *common lower bound*. This implies that $\mathrm{PF}_{d_{\mathrm{TV}}}$ can be considered as a generalization of RDP.

**Theorem 1.** *The Group Precision and Group Recall of any restoration method satisfy*

$$GP(a) \geq 1 - GPI_{d_{TV}}(a), \tag{13}$$

$$GR(a) \geq 1 - GPI_{d_{TV}}(a), \tag{14}$$

*for all $a \in \mathrm{supp}\, p_A$.*

Although using $d_{\text{TV}}(\cdot, \cdot)$ provides a straightforward relationship between $\text{PF}_{d_{\text{TV}}}$ and RDP, other types of divergences may not necessarily indicate GP and GR so explicitly. The perceptual quality & variation of a group's reconstructions may be defined in many different ways [71], and the GPI implicitly entangles these two desired properties.

The mathematical notations and fairness definitions are summarized in Appendix A. To further develop our understanding of PF, the next section presents several introductory theorems.

## 3 Theoretical results

Image restoration algorithms can generally be categorized into three groups: (1) Algorithms targeting the best possible average distortion (*e.g.*, good PSNR) [3, 21, 47, 48, 83, 85, 97–100], (2) algorithms that strive to achieve good average distortion but prioritize attaining best PI [1, 19, 27, 47, 61, 80, 83–85, 88, 89, 93, 100, 104], and (3) algorithms attempting to sample from the posterior distribution $p_{X|Y}$ of the given task at hand [16, 40–42, 51, 58, 76, 86, 91]. In Appendix B, we demonstrate on a simple toy example that all these types of algorithms may achieve poor PF, implying that perfect PF is not a property that can be obtained trivially. Namely, even when using common reconstruction algorithms such as the Minimum Mean-Squared-Error (MMSE) estimator or the posterior sampler, one group may attain far worse GPI than another group. It is therefore tempting to ask in which scenarios there exists an algorithm capable of achieving perfect GPI for all groups simultaneously. As stated in the following theorem, this desired property is unattainable when the degradation is sufficiently severe.

**Theorem 2.** *Suppose that* $\exists a_1, a_2 \in \text{supp}\, p_A$ *such that*

$$\mathbb{P}(X \in \mathcal{X}_{a_1} \cap \mathcal{X}_{a_2} | A = a_i) < \mathbb{P}(Y \in \mathcal{Y}_{a_1} \cap \mathcal{Y}_{a_2} | A = a_i), \tag{15}$$

*for both* $i = 1, 2$, *where* $\mathcal{X}_{a_i} = \text{supp}\, p_{X|A}(\cdot|a_i)$ *and* $\mathcal{Y}_{a_i} = \text{supp}\, p_{Y|A}(\cdot|a_i)$. *Then,* $GPI_d(a_1)$ *and* $GPI_d(a_2)$ *cannot both be equal to zero.*

In words, Theorem 2 states that when the degraded images of different groups are "more overlapping" than their ground truth images, at least one group must have sub-optimal GPI. Importantly, note that perfect GPI can always be achieved for some group corresponding to $A = a$ individually, by ignoring the input and sampling from $p_{X|A}(\cdot|a)$. Hence, Theorem 2 implies that, for sufficiently severe degradations, one may attempt to approach zero GPI for all groups simultaneously, until the GPI of one group hinders that of another one. But what about algorithms that just attain perfect *overall* PI? Can such algorithms also attain perfect PF? As stated in the following theorem, it turns out that these two desired properties (perfect PI and perfect PF) are often incompatible.

**Theorem 3.** *Suppose that* $A$ *takes discrete values,* $\hat{X}$ *attains perfect* $PI_d$ *(*$p_{\hat{X}} = p_X$*), and* $\exists a, a_m \in \text{supp}\, p_A$ *such that* $GPI_d(a) > 0$ *and* $\mathbb{P}(A = a_m) > 0.5$. *Then,* $\hat{X}$ *cannot achieve perfect* $PF_{d_{\text{TV}}}$.

In words, when there exists a majority group in the data distribution, Theorem 3 states that an algorithm with perfect PI, whose GPI is not perfect *even for only one group*, cannot achieve perfect $\text{PF}_{d_{\text{TV}}}$. This intriguing outcome results from the following convenient relationship between the GPIs of different groups for algorithms with perfect PI.

**Theorem 4.** *Suppose that* $A$ *takes discrete values and* $\hat{X}$ *attains perfect* $PI_d$ *(*$p_{\hat{X}} = p_X$*). Then,*

$$GPI_{d_{\text{TV}}}(a) \leq \frac{1}{\mathbb{P}(A = a)} \sum_{a' \neq a} \mathbb{P}(A = a') GPI_{d_{\text{TV}}}(a') \tag{16}$$

*for every* $a$ *with* $\mathbb{P}(A = a) > 0$.

This theorem is, perhaps, counter-intuitive. Indeed, for algorithms with perfect PI, improving the $\text{GPI}_{d_{\text{TV}}}$ of one group can only *improve* the $\text{GPI}_{d_{\text{TV}}}$ of other groups, and this is true *even if the groups do not overlap*[1]. While this may seem contradictory to Theorem 2, note that such a relationship holds until the algorithm can no longer attain perfect PI. The example in Appendix B demonstrates this theorem.

---

[1]Two groups with attributes $a_1, a_2$ are overlapping if $\mathbb{P}(X \in \mathcal{X}_{a_1} \cap \mathcal{X}_{a_2}) > 0$, where $\mathcal{X}_{a_i} = \text{supp}\, p_{X|A}(\cdot|a_i)$.

# 4 Experiments

We demonstrate the superiority of PF over RDP in detecting fairness bias in face image super-resolution. Our analysis considers various aspects, including different types of degradations, and fairness evaluations across four groups categorized by ethnicity and age. First, we show that RDP incorrectly attributes fairness in a simple scenario where fairness is clearly violated. In contrast, PF successfully detects the bias. Second, we showcase a scenario where PF uncovers potential malicious intent. Specifically, it can detect bias injected into the system via adversarial attacks, a situation again missed by RDP.

## 4.1 Synthetic data sets

In the following sections we assess the fairness of leading face image restoration methods through the lens of PF and RDP. Such methods are often trained and evaluated on high-quality, aligned face image datasets like CelebA-HQ [36] and FFHQ [37], which lack ground truth labels for sensitive attributes such as ethnicity. Moreover, these datasets are prone to inherent biases, *e.g.*, they contain very few images for certain demographic groups [31, 35, 69], and it is unclear whether images from different groups have similar levels of image quality and variation (prior work suggests that they might not [11]). To address these limitations, we leverage an image-to-image translation model that takes a text instruction as additional input. This model allows us to generate four synthetic fairness groups with high-quality, aligned face images. Specifically, we translate each image $x$ from the CelebA-HQ [36] test partition into four different images representing Asian/non-Asian and young/old individuals[2]. We use a unique text instruction for each translation. For example, the text instruction ''120 years old human, Asian, natural image, sharp, DSLR'' translates $x$ into an image of an old&Asian individual. Finally, we include each resulting image in its corresponding group data only if *all* translations are successful according to the FairFace combined age & ethnicity classifier [35]. This involves classifying the ethnicity and age of the translated images and ensuring that old individuals are categorized as 70+ years old, young individuals are categorized as any other age group, Asian individuals are classified as either Southeast or East Asian, and non-Asian individuals are classified as belonging to any other ethnicity group. See Appendix G.1 for more details and for the visualization of the results.

**Disclaimer.** Importantly, we note that the generated synthetic data sets may impose offensive biases and stereotypes. We use such data sets solely to investigate the fairness of image restoration methods and verify the practical utility of our work. We do not intend to discriminate against any identity group or cultures in any way.

## 4.2 Perceptual Fairness vs. Representation Demographic Parity

We consider several image super-resolution tasks using the average-pooling down-sampling operator with scale factors $s \in \{4, 8, 16, 32\}$, and statistically independent additive white Gaussian noise of standard deviation $\sigma_N \in \{0, 0.1, 0.25\}$. In Appendix I we also conduct experiments on image denoising and deblurring. The algorithms DDNM$^+$ [86], DDRM [42], DPS [16], and PiGDM [76] are evaluated on all scale factors, and GFPGAN [84], VQFR [27], GPEN [93], DiffBIR [49], Code-Former [104], RestoreFormer++ [89], and RestoreFormer [88] are evaluated only on the $\times 4$ and $\times 8$ scale factors (these algorithms produce completely wrong outputs for the other scale factors). To assess the PF of each algorithm, we compute the GPI$_{\text{KID}}$ of each group using the Kernel Inception Distance (KID) [9] and the features extracted from the last pooling layer of the FairFace combined age & ethnicity classifier [35]. In Appendix G.4 we utilize the Fréchet Inception Distance (FID) [30] instead of KID, and in Appendix G.5 we assess other types of group metrics such as PSNR. Additionally, we provide in Appendix G.6 an ablation study of alternative feature extractors. To assess RDP, we use the same FairFace classifier to compute the GP of each group. As done in [34], we approximate the GP of each group by the classification hit rate, which is the ratio between the number of the group's reconstructions that are classified as belonging to the group and the total number of the group's inputs. Qualitative and quantitative results for $s = 32, \sigma_N = 0.0$ are presented in Figure 2.

---

[2]We choose to consider these fairness groups since image restoration algorithms are likely biased towards young and white demographics, given the overrepresentation of such groups in common training datasets (*e.g.*, FFHQ, CelebA). Namely, groups of Asian and/or old individuals are typically underrepresented in such datasets.

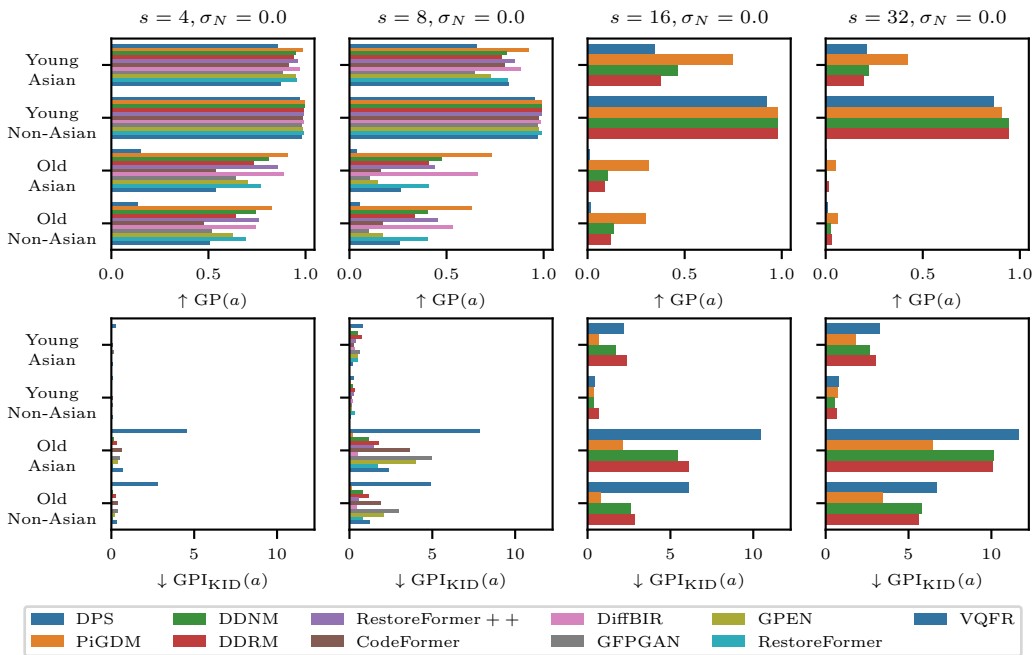

**Figure 3:** Comparison of the GP and the $\text{GPI}_{\text{KID}}$ of different fairness groups, using various state-of-the-art face image super-resolution methods. In most experiments, $\text{GPI}_{\text{KID}}$ suggests a fairness discrepancy between the groups old&non-Asian and old&Asian, while the GP of these groups is roughly equal.

Quantitative results for all values of $s$ and $\sigma_N = 0.0$ are shown in Figure 3. Complementary details and results are provided in Appendix G.

Figure 3 shows that the group young&non-Asian receives the best overall treatment in terms of both GP and $\text{GPI}_{\text{KID}}$. This result is not surprising, since the training data sets of the evaluated algorithms (*e.g.*, FFHQ) are known to be biased towards young and white demographics [50, 63]. However, while most algorithms appear to treat the groups old&Asian and old&non-Asian quite similarly in terms of GP, the $\text{GPI}_{\text{KID}}$ indicates a clear disadvantage for the former group. Indeed, by examining ethnicity and age separately using the FairFace classifier, we show in Appendix G.7 that, according to RDP, the group old&non-Asian exhibits better preservation of the ethnicity attribute compared to the group old&Asian, while the age attribute remains equally preserved for both groups. This highlights that RDP is *strongly* dependent on the granularity of the fairness groups (as suggested in [34]), since slightly altering the groups' partitioning may *completely* obscure the fact that an algorithm treats certain attributes more favorably than others. However, as our results show, this issue is alleviated when adopting $\text{GPI}_{\text{KID}}$ instead of GP. Namely, the ethnicity bias is still detected by comparing the $\text{GPI}_{\text{KID}}$ of different groups, even though the fairness groups are partitioned based on age and ethnicity combined.

### 4.3 Adversarial bias detection

In Section 2.2.1 we discussed the limitations of fairness definitions such as RDP. For instance, an algorithm might satisfy RDP by always generating the same output for degraded images of a particular group, even if it produces perfect results for another. However, such an extreme scenario is not common in practice. Indeed, real-world imaging systems often involve degradations that are not too severe, and well-trained algorithms perform impressively well when applied to different groups (see, *e.g.*, Figure 4b). So what practical advantage does PF have over RDP in such circumstances? Here, we demonstrate that a malicious user can manipulate the facial features (*e.g.*, wrinkles) of a group's reconstructions without violating fairness according to RDP, but violating fairness according to PF. In particular, we consider only the ethnicity sensitive attribute by taking the young&Asian group as Asian, and the young&non-Asian group as non-Asian. Then, we use the RestoreFormer++ method, which roughly satisfies RDP with respect to these groups (see Figure 4a, where GP is evaluated by

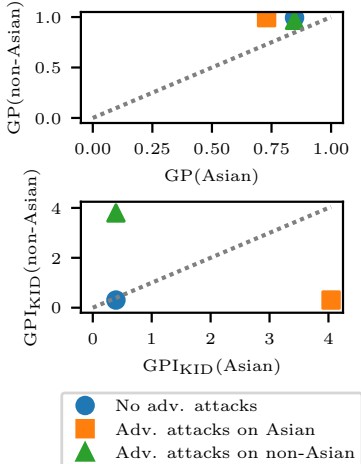

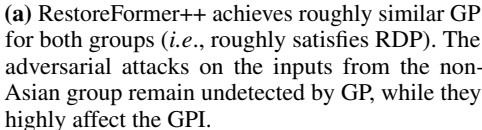

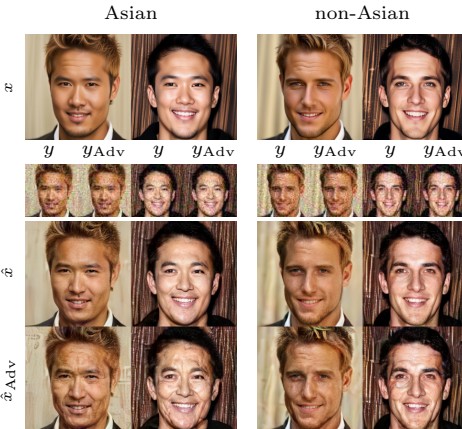

**(a)** RestoreFormer++ achieves roughly similar GP for both groups (*i.e.*, roughly satisfies RDP). The adversarial attacks on the inputs from the non-Asian group remain undetected by GP, while they highly affect the GPI.

**(b)** Visual results. $y$ and $x$ are the original input and the source image, respectively. $y_{\text{Adv}}$ and $\hat{x}_{\text{Adv}}$ are the adversarial input and its corresponding output, respectively. Each $y_{\text{Adv}}$ successfully alters the output facial features. Indeed, $\hat{x}_{\text{Adv}}$ clearly contains a face with more wrinkles than $x$.

**Figure 4:** Using adversarial attacks to inject bias into the outputs of RestoreFormer++, in a setting where it (roughly) satisfies RDP. Such attacks are detected by PF but not by RDP.

classifying ethnicity alone), and perform adversarial attacks on the inputs of each group to manipulate the outputs such that they are classified as belonging to the 70+ age category. The fact that the GP of each group is quite large implies that the malicious user can classify ethnicity quite accurately from the degraded images, and then manipulate the inputs only for the group it wishes to harm (we skip such a classification step and simply attack all of the group's inputs). Such attacks are anticipated to succeed due to the perception-robustness tradeoff [59, 60]. Complementary details of this experiment are provided in Appendix H.

In Figure 4, we present both quantitative and qualitative results demonstrating that the attacks on the non-Asian group are not detected by RDP. However, we clearly observe that these attacks are successfully identified by the $\text{GPI}_{\text{KID}}$ of each group. This again highlights that PF is less sensitive to the choice (partitioning) of fairness groups compared to RDP. Specifically, age must be considered as a sensitive attribute to detect such a bias via RDP. Yet, even then, the malicious user may still inject other types of biases. Conversely, PF does not suffer from this limitation, as any attempt to manipulate the distribution of a group's reconstructions would be reflected in the group's GPI.

## 5 Discussion

Different demographic groups can utilize an image restoration algorithm, and fairness in this context asserts whether the algorithm "treats" all groups equally well. In this paper, we introduce the notion of Perceptual Fairness (PF) to assess whether such a desired property is upheld. We delve into the theoretical foundation of PF, demonstrate its practical utility, and discuss its superiority over existing fairness definitions. Still, our work is not without limitations. First, while PF alleviates the strong dependence of RDP on the choice of fairness groups [34] (as demonstrated in Section 4), it still cannot guarantee fairness for any arbitrary group partitioning simultaneously (a property referred to as *obliviousness* in [34]). Second, our current theorems are preliminary, requiring further research to fully understand the nature of PF. For example, the severity of the tradeoff between the GPI scores of different groups (Theorem 2) and that of the tradeoff between PF and PI (Theorem 3) remain unclear. Third, we do not address the nature of optimal estimators that achieve good or perfect PF. What is their best possible distortion (*e.g.*, MSE) and best possible PI? Fourth, on the practical side, we show in Appendix G.6 that effectively evaluating PF using metrics such as KID necessitates utilizing image

features extracted from a classifier dedicated to handling the considered sensitive attributes (*e.g.*, an age and ethnicity classifier). However, this is not a disadvantage compared to previous fairness notions (RDP, CPR and PR), which also require such a classifier. Lastly, while the proposed GPI may be suitable for evaluating fairness in general-content natural images, we considered only human face images due to their societal implications, namely since fairness issues are particularly critical when dealing with such images. For example, if a general-content image restoration algorithm performs better on images with complex structures than on images of clear skies, this discrepancy is unlikely to be problematic for practitioners, as long as the algorithm attains good performance overall. Moreover, previous works [34] evaluated fairness with respect to non-human subjects (*e.g.*, dogs and cats), but these studies provide limited insights into human-related fairness issues, which often arise due to subtle differences between images (*e.g.*, wrinkles). Expanding our method to other datasets remains an avenue for future work.

## 6   Societal impact

Designing fair and unbiased image restoration algorithms is critical for various AI applications and downstream tasks that rely on them, such as facial recognition, image classification, and image editing. By proposing practically useful and well-justified fairness definitions, we can detect (and mitigate) bias in these tasks, ultimately leading to fairer societal outcomes. This fosters increased trust and adoption of AI technology, contributing to a more equitable and responsible use of AI in society.

## Acknowledgments

This research was partially supported by the Israel Science Foundation (ISF) under Grant 2318/22 and by the Council For Higher Education - Planning & Budgeting Committee.

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

| Name / Notation | Meaning / Formal definition |
|---|---|
| $X$ | Ground truth image (a random vector) |
| $Y$ | Degraded measurement (a random vector) |
| $\hat{X}$ | Reconstructed image (a random vector) |
| $p_X$ | P.d.f of the ground truth images |
| $p_Y$ | P.d.f of the degraded measurements |
| $p_{\hat{X}}$ | P.d.f of the reconstructed images |
| Perceptual Index ($\text{PI}_d$ or PI) | $d(p_X, p_{\hat{X}})$ |
| $A$ | Sensitive attribute (a random vector) |
| $p_{X|A}(\cdot|a)$ | P.d.f of the ground truth images of $A = a$ |
| $p_{Y|A}(\cdot|a)$ | P.d.f of the degraded measurements of $A = a$ |
| $p_{\hat{X}|A}(\cdot|a)$ | P.d.f of the reconstructed images of $A = a$ |
| $\mathcal{X}_a$ | $\operatorname{supp} p_{X|A}(\cdot|a)$ |
| $\mathcal{Y}_a$ | $\operatorname{supp} p_{Y|A}(\cdot|a)$ |
| $\hat{\mathcal{X}}_a$ | $\operatorname{supp} p_{\hat{X}|A}(\cdot|a)$ |
| Group Perceptual Index ($\text{GPI}_d(a)$, $\text{GPI}_d$, or GPI) | $d(p_{X|A}(\cdot|a), p_{\hat{X}|A}(\cdot|a))$ |
| Group Precision ($\text{GP}(a)$ or GP) | $\mathbb{P}(\hat{X} \in \mathcal{X}_a|A = a)$ |
| Group Recall ($\text{GR}(a)$ or GR) | $\mathbb{P}(X \in \hat{\mathcal{X}}_a|A = a)$ |
| Representation Demographic Parity (RDP) | $\forall a_1, a_2 : \text{GP}(a_1) = \text{GP}(a_2)$ |
| Proportional Representation (PR) | $\forall a : \mathbb{P}(X \in \mathcal{X}_a) = \mathbb{P}(\hat{X} \in \mathcal{X}_a)$ |
| Conditional Proportional Representation (CPR) | $\forall a, y : \mathbb{P}(X \in \mathcal{X}_a|Y = y) = \mathbb{P}(\hat{X} \in \mathcal{X}_a|Y = y)$ |
| Perceptual Fairness ($\text{PF}_d$ or PF) | $\forall a_1, a_2 : \text{GPI}_d(a_1) = \text{GPI}_d(a_2)$ |

**Table 1:** Summary of mathematical notations and fairness definitions used in this paper.

## A   Summary of mathematical notations and fairness definitions

We summarize in Table 1 the mathematical notations and fairness definitions used in this paper.

## B   Toy signal restoration example

The following toy signal restoration example demonstrates that common estimators (*e.g.*, the stochastic estimator which samples from the posterior distribution $p_{X|Y}$) do not trivially achieve perfect PF.

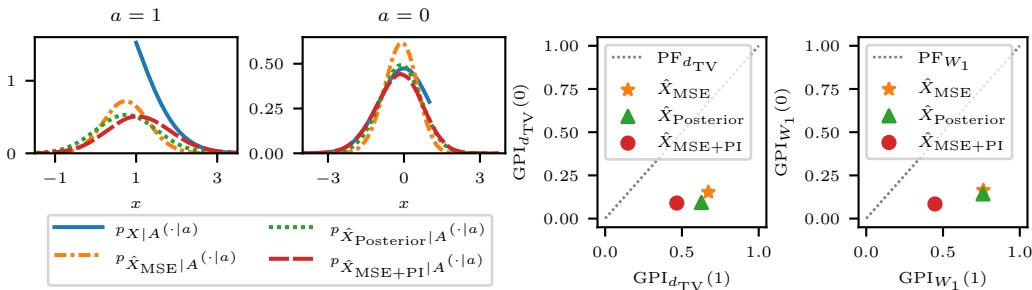

**Figure 5:** Illustration of Example 1. **Left**: Conditional probability density functions $p_{X|A}(\cdot|a)$, $p_{\hat{X}_{\text{MSE}}|A}(\cdot|a)$, $p_{\hat{X}_{\text{Posterior}}|A}(\cdot|a)$, and $p_{\hat{X}_{\text{MSE+PI}}|A}(\cdot|a)$, where $a = 1$ (left plot) or $a = 0$ (right plot). **Right**: The $\text{GPI}_{d_{\text{TV}}}$ and $\text{GPI}_{W_1}$ of each group (associated with $a = 1$ or $a = 0$). The dotted lines $\text{PF}_{d_{\text{TV}}}$ or $\text{PF}_{W_1}$ correspond to the points where perfect $\text{PF}_{d_{\text{TV}}}$ or perfect $\text{PF}_{W_1}$ is achieved, respectively. It is clear that all three estimators achieve sub-optimal $\text{PF}_{d_{\text{TV}}}$ and sub-optimal $\text{PF}_{W_1}$. See Appendix B for more details.

**Example 1.** *Suppose that $X, N \sim \mathcal{N}(0, 1)$ are statistically independent random variables, and let $Y = X + N$. In this case, it is known that $\hat{X}_{MSE} = \frac{1}{2}Y$ is the estimator that attains the lowest possible Mean-Squared-Error (MSE), $\hat{X}_{Posterior} = \frac{1}{2}Y + W$ where $W \sim \mathcal{N}(0, \frac{1}{2})$ is statistically independent of $X$ and $Y$, is the estimator that samples from the posterior distribution $p_{X|Y}$, and $\hat{X}_{MSE+PI} = \frac{1}{\sqrt{2}}Y$*

*is the estimator that attains the lowest possible MSE among all estimators that satisfy $p_{\hat{X}} = p_X$ (perfect PI$_d$) [10, 23]. Now, consider the "sensitive attribute" $A = \mathbb{1}_{X \geq 1}$. All of these commonly used estimators produce much better (lower) GPI$_{d_{TV}}$ and GPI$_{W_1}$ for the group associated with $A = 0$, which, in this case, is a majority satisfying $\mathbb{P}(A = 0) \approx 0.8413$ (see Figure 5).*

### B.1 Conditional density plots

The density $p_{X|A}(x|a)$ is obtained using the closed form solution of a truncated normal distribution,

$$p_{X|A}(x|1) = \frac{\phi(x)}{\Phi(\infty) - \Phi(1)}, \tag{17}$$

$$p_{X|A}(x|0) = \frac{\phi(x)}{\Phi(1) - \Phi(-\infty)}, \tag{18}$$

where $\phi(x)$ is a normal density and $\Phi(x)$ is its cumulative distribution,

$$\phi(x) = \frac{1}{\sqrt{2\pi}} e^{-\frac{1}{2}x^2}, \tag{19}$$

$$\Phi(x) = \frac{1}{2}\left(1 + \mathrm{erf}\left(\frac{x}{\sqrt{2}}\right)\right), \tag{20}$$

and $p_{X|A}(x|1) = 0$ and $p_{X|A}(x|0) = 0$ for every $x \geq 1$ and $x \leq 1$, respectively. The densities $p_{\hat{X}_{\mathrm{MSE}}|A}(\cdot|a), p_{\hat{X}_{\mathrm{MSE+PQ}}|A}(\cdot|a)$ and $p_{\hat{X}_{\mathrm{Posterior}}|A}(\cdot|a)$ are obtained by feeding these algorithms with the degraded measurements corresponding to $X \geq 1$ (for $a = 1$) and to $X < 1$ (for $a = 0$), separately. This is achieved by generating samples $x \sim p_X$ and $y \sim p_{Y|X}(\cdot|x)$, and then partitioning these samples into two sets of measurements based on the value of $x$. We then perform Kernel Density Estimation (KDE) [64] on the reconstructions of each group to obtain their density, using the function `seaborn.kdeplot` [90] with the arguments `bw_adjust=2`, `common_norm=False`, `gridsize=200`. The number of samples used to compute the KDE is set to 200,000 for both $a = 1$ and $a = 0$.

### B.2 Computation of the total variation distance $d_{\mathbf{TV}}$ and of the Wasserstein distance $W_1$

The value of GPI$_{d_{\mathrm{TV}}}(a)$ for a given algorithm $\hat{X}$ is defined by the total variation distance

$$\mathrm{GPI}_{d_{\mathrm{TV}}}(a) = d_{\mathrm{TV}}(p_{X|A}(\cdot|a), p_{\hat{X}|A}(\cdot|a)) = \frac{1}{2}\int \left|p_{X|A}(x|a) - p_{\hat{X}|A}(x|a)\right| dx. \tag{21}$$

To compute this integral, we use the function `scipy.integrate.quad` [82] with parameters (`a=-1000, b=1000, limit=500, points=[1.0]`). At each point $x$, the integrand

$$\left|p_{X|A}(x|a) - p_{\hat{X}|A}(x|a)\right| \tag{22}$$

is evaluated using the closed form solution of $p_{X|A}(\cdot|a)$ and the pre-computed KDE density of each $p_{\hat{X}|A}(\cdot|a)$.

The value of GPI$_{W_1}(a)$ for a given algorithm $\hat{X}$ is the Wasserstein 1-distance between $p_{X|A}(\cdot|a)$ and $p_{\hat{X}|A}(\cdot|a)$. To approximate this distance, we utilize the function `scipy.stats.wasserstein_distance` with the previously obtained 200,000 samples from $p_{X|A}(\cdot|a)$ and 200,000 samples from $p_{\hat{X}|A}(\cdot|a)$.

## C Proof of Theorem 1

**Theorem 1.** *The Group Precision and Group Recall of any restoration method satisfy*

$$GP(a) \geq 1 - GPI_{d_{TV}}(a), \tag{13}$$

$$GR(a) \geq 1 - GPI_{d_{TV}}(a), \tag{14}$$

*for all $a \in \mathrm{supp}\, p_A$.*

*Proof.* For every $a, x$, it holds that

$$p_{\hat{X}|A}(x|a) \geq \min\left\{p_{X|A}(x|a), p_{\hat{X}|A}(x|a)\right\}. \tag{23}$$

Moreover, the value of $\min\left\{p_{X|A}(x|a), p_{\hat{X}|A}(x|a)\right\}$ is zero for every $x \notin \operatorname{supp} p_{X|A}(\cdot|a)$, so

$$\int_{\operatorname{supp} p_{X|A}(\cdot|a)} \min\left\{p_{X|A}(x|a), p_{\hat{X}|A}(x|a)\right\}dx = \int \min\left\{p_{X|A}(x|a), p_{\hat{X}|A}(x|a)\right\}dx. \tag{24}$$

Thus,

$$\mathrm{GP}(a) = \mathbb{P}(\hat{X} \in \mathcal{X}_a | A = a) \tag{25}$$

$$= \mathbb{P}(\hat{X} \in \operatorname{supp} p_{X|A}(\cdot|a)|A = a) \tag{26}$$

$$= \int_{\operatorname{supp} p_{X|A}(\cdot|a)} p_{\hat{X}|A}(x|a)dx \tag{27}$$

$$\geq \int_{\operatorname{supp} p_{X|A}(\cdot|a)} \min\left\{p_{X|A}(x|a), p_{\hat{X}|A}(x|a)\right\}dx \tag{28}$$

$$= \int \min\left\{p_{X|A}(x|a), p_{\hat{X}|A}(x|a)\right\}dx \tag{29}$$

$$= \int \frac{1}{2}\left(p_{\hat{X}|A}(x|a) + p_{X|A}(x|a) - \left|p_{\hat{X}|A}(x|a) - p_{X|A}(x|a)\right|\right)dx \tag{30}$$

$$= \frac{1}{2}\int \left(p_{\hat{X}|A}(x|a) + p_{X|A}(x|a)\right)dx - \frac{1}{2}\int \left|p_{\hat{X}|A}(x|a) - p_{X|A}(x|a)\right|dx \tag{31}$$

$$= 1 - d_{\mathrm{TV}}(p_{X|A}(\cdot|a), p_{\hat{X}|A}(\cdot|a)) \tag{32}$$

$$= 1 - \mathrm{GPI}_{d_{\mathrm{TV}}}(a). \tag{33}$$

By replacing the roles of $p_{\hat{X}|A}(x|a)$ and $p_{X|A}(x|a)$, the result $\mathrm{GR}(a) \geq 1 - \mathrm{GPI}_{d_{\mathrm{TV}}}(a)$ can be derived with identical steps using the same mathematical arguments. $\square$

## D   Proof of Theorem 2

**Theorem 2.** *Suppose that $\exists a_1, a_2 \in \operatorname{supp} p_A$ such that*

$$\mathbb{P}(X \in \mathcal{X}_{a_1} \cap \mathcal{X}_{a_2} | A = a_i) < \mathbb{P}(Y \in \mathcal{Y}_{a_1} \cap \mathcal{Y}_{a_2} | A = a_i), \tag{15}$$

*for both $i = 1, 2$, where $\mathcal{X}_{a_i} = \operatorname{supp} p_{X|A}(\cdot|a_i)$ and $\mathcal{Y}_{a_i} = \operatorname{supp} p_{Y|A}(\cdot|a_i)$. Then, $\mathrm{GPI}_d(a_1)$ and $\mathrm{GPI}_d(a_2)$ cannot both be equal to zero.*

*Proof.* Suppose by contradiction that $p_{\hat{X}|A}(\cdot|a_i) = p_{X|A}(\cdot|a_i)$ for both $i = 1, 2$. Thus,

$$1 = \mathbb{P}(X \in \mathcal{X}_{a_i} | A = a_i) \tag{34}$$

$$= \mathbb{P}(\hat{X} \in \mathcal{X}_{a_i} | A = a_i) \tag{35}$$

$$= \int_{\mathcal{X}_{a_i}} p_{\hat{X}|A}(x|a_i)dx \tag{36}$$

$$= \int\int_{\mathcal{X}_{a_i}} p_{\hat{X},Y|A}(x|a_i)dxdy \tag{37}$$

$$= \int\int_{\mathcal{X}_{a_i}} p_{\hat{X}|A,Y}(x|a_i)p_{Y|A}(y|a_i)dxdy \tag{38}$$

$$= \int_{\mathcal{Y}_{a_i}}\int_{\mathcal{X}_{a_i}} p_{\hat{X}|Y}(x|y)p_{Y|A}(y|a_i)dxdy \tag{39}$$

$$= \int_{\mathcal{Y}_{a_i}} p_{Y|A}(y|a_i)\left(\int_{\mathcal{X}_{a_i}} p_{\hat{X}|Y}(x|y)dx\right)dy \tag{40}$$

$$= \int_{\mathcal{Y}_{a_i}} p_{Y|A}(y|a_i)\mathbb{P}(\hat{X} \in \mathcal{X}_{a_i}|Y = y)dy, \tag{41}$$

where Equation (39) holds from the assumption that $A$ and $\hat{X}$ are statistically independent given $Y$, and from the fact that $p_{Y|A}(y|a_i) = 0$ for every $y \notin \mathcal{Y}_{a_i}$. We will show that $\mathbb{P}(\hat{X} \in \mathcal{X}_{a_i}|Y = y) = 1$ for almost every $y \in \mathcal{Y}_{a_i}$. Indeed, if this does not hold, then for some $\mathcal{T}_i \subseteq \mathcal{Y}_{a_i}$ with $\mathbb{P}(Y \in \mathcal{T}_i|A = a_i) > 0$ we have $\mathbb{P}(\hat{X} \in \mathcal{X}_{a_i}|Y = y) < 1$ for every $y \in \mathcal{T}_i$. Thus,

$$1 = \int_{\mathcal{Y}_{a_i}} p_{Y|A}(y|a_i)\mathbb{P}(\hat{X} \in \mathcal{X}_{a_i}|Y = y)dy \tag{42}$$

$$= \int_{\mathcal{Y}_{a_i}\setminus\mathcal{T}_i} p_{Y|A}(y|a_i)\mathbb{P}(\hat{X} \in \mathcal{X}_{a_i}|Y = y)dy + \int_{\mathcal{T}_i} p_{Y|A}(y|a_i)\mathbb{P}(\hat{X} \in \mathcal{X}_{a_i}|Y = y)dy \tag{43}$$

$$< \int_{\mathcal{Y}_{a_i}\setminus\mathcal{T}_i} p_{Y|A}(y|a_i)\mathbb{P}(\hat{X} \in \mathcal{X}_{a_i}|Y = y)dy + \int_{\mathcal{T}_i} p_{Y|A}(y|a_i)dy \tag{44}$$

$$\leq \int_{\mathcal{Y}_{a_i}\setminus\mathcal{T}_i} p_{Y|A}(y|a_i)dy + \int_{\mathcal{T}_i} p_{Y|A}(y|a_i)dy \tag{45}$$

$$= \int_{\mathcal{Y}_{a_i}} p_{Y|A}(y|a_i)dy \tag{46}$$

$$= 1, \tag{47}$$

which is not possible. So, $\mathbb{P}(\hat{X} \in \mathcal{X}_{a_i}|Y = y) = 1$ for almost every $y \in \mathcal{Y}_{a_i}$. Now, from basic rules of probability theory, we have

$$\mathbb{P}(X \in \mathcal{X}_{a_1} \cap \mathcal{X}_{a_2}|A = a_1) = \mathbb{P}(X \in \mathcal{X}_{a_1}|A = a_1)$$
$$+ \mathbb{P}(X \in \mathcal{X}_{a_2}|A = a_1)$$
$$- \mathbb{P}(X \in \mathcal{X}_{a_1} \cup \mathcal{X}_{a_2}|A = a_1), \tag{48}$$

where the first and last terms on the right hand side cancel out (from the definition of $\mathcal{X}_{a_1}$, they are both equal to 1). Thus, we have

$$\mathbb{P}(X \in \mathcal{X}_{a_1} \cap \mathcal{X}_{a_2}|A = a_1) = \mathbb{P}(X \in \mathcal{X}_{a_2}|A = a_1), \tag{49}$$

and finally,

$$\mathbb{P}(X \in \mathcal{X}_{a_1} \cap \mathcal{X}_{a_2}|A = a_1) = \mathbb{P}(X \in \mathcal{X}_{a_2}|A = a_1) \tag{50}$$

$$= \mathbb{P}(\hat{X} \in \mathcal{X}_{a_2}|A = a_1) \tag{51}$$

$$= \int_{\mathcal{Y}_{a_1}} p_{Y|A}(y|a_1)\mathbb{P}(\hat{X} \in \mathcal{X}_{a_2}|Y = y)dy \tag{52}$$

$$\geq \int_{\mathcal{Y}_{a_1}\cap\mathcal{Y}_{a_2}} p_{Y|A}(y|a_1)\mathbb{P}(\hat{X} \in \mathcal{X}_{a_2}|Y = y)dy \tag{53}$$

$$= \int_{\mathcal{Y}_{a_1}\cap\mathcal{Y}_{a_2}} p_{Y|A}(y|a_1)dy \tag{54}$$

$$= \mathbb{P}(Y \in \mathcal{Y}_{a_1} \cap \mathcal{Y}_{a_2}|A = a_1), \tag{55}$$

where Equation (51) follows from the contradictory assumption that $p_{\hat{X}|A}(\cdot|a_i) = p_{X|A}(\cdot|a_i)$, Equation (52) follows from the same steps that led to Equation (41), and Equation (54) follows from our previous finding that $\mathbb{P}(\hat{X} \in \mathcal{X}_{a_i}|Y = y) = 1$ for every $y \in \mathcal{Y}_{a_i}$ (we have $y \in \mathcal{Y}_{a_1}\cap\mathcal{Y}_{a_2}$ in the integrand, so $y \in \mathcal{Y}_{a_2}$). However, it is given that $\mathbb{P}(X \in \mathcal{X}_{a_1} \cap \mathcal{X}_{a_2}|A = a_1) < \mathbb{P}(Y \in \mathcal{Y}_{a_1} \cap \mathcal{Y}_{a_2}|A = a_1)$, so we have established a contradiction. $\square$

## E  Proof of Theorem 3

**Theorem 3.** *Suppose that $A$ takes discrete values, $\hat{X}$ attains perfect $PI_d$ ($p_{\hat{X}} = p_X$), and $\exists a, a_m \in$ supp $p_A$ such that $GPI_d(a) > 0$ and $\mathbb{P}(A = a_m) > 0.5$. Then, $\hat{X}$ cannot achieve perfect $PF_{d_{TV}}$.*

*Proof.* Suppose that $GPI_{d_{TV}}(a_m) = 0$. From the assumptions, there exists $a \neq a_m$ such that $GPI_d(a) > 0$, so $GPI_{d_{TV}}(a) > 0$. This means that $PF_{d_{TV}}$ is not perfect.

Otherwise, suppose that $\text{GPI}_{d_{\text{TV}}}(a_m) > 0$. Thus, from Theorem 4 we have

$$\text{GPI}_{d_{\text{TV}}}(a_m) \leq \frac{1 - \mathbb{P}(A = a_m)}{\mathbb{P}(A = a_m)} \max_{a' \neq a_m} \text{GPI}_{d_{\text{TV}}}(a') \tag{56}$$

$$< \max_{a' \neq a_m} \text{GPI}_{d_{\text{TV}}}(a) \tag{57}$$

$$= \text{GPI}_{d_{\text{TV}}}(a^*), \tag{58}$$

where Equation (57) holds since $\frac{1 - \mathbb{P}(A = a_m)}{\mathbb{P}(A = a_m)} < 1$, and Equation (58) holds by defining

$$a^* = \arg\max_{a' \neq a} \text{GPI}_{d_{\text{TV}}}(a'). \tag{59}$$

Thus, we have found two groups $a_m$ and $a^*$ such that $\text{GPI}_{d_{\text{TV}}}(a_m) < \text{GPI}_{d_{\text{TV}}}(a^*)$, so $\text{PF}_{d_{\text{TV}}}$ cannot be perfect. $\qquad\square$

# F    Proof of Theorem 4

**Theorem 4.** *Suppose that $A$ takes discrete values and $\hat{X}$ attains perfect $PI_d$ ($p_{\hat{X}} = p_X$). Then,*

$$GPI_{d_{\text{TV}}}(a) \leq \frac{1}{\mathbb{P}(A = a)} \sum_{a' \neq a} \mathbb{P}(A = a') GPI_{d_{\text{TV}}}(a') \tag{16}$$

*for every $a$ with $\mathbb{P}(A = a) > 0$.*

*Proof.* For every $a$, let us denote $P_a = \mathbb{P}(A = a)$. Suppose that $\hat{X}$ attains perfect perceptual index, so $p_{\hat{X}} = p_X$. From the marginalization of probability density functions, it holds that

$$p_X(x) = \sum_a P_a p_{X|A}(x|a), \tag{60}$$

$$p_{\hat{X}}(x) = \sum_a P_a p_{\hat{X}|A}(x|a), \tag{61}$$

and since $p_{\hat{X}} = p_X$ we have

$$\sum_a P_a p_{X|A}(x|a) = \sum_a P_a p_{\hat{X}|A}(x|a). \tag{62}$$

Let $a$ be some group with $P_a > 0$. By rearranging Equation (62) we get

$$P_a(p_{X|A}(x|a) - p_{\hat{X}|A}(x|a)) = \sum_{a' \neq a} P_{a'}(p_{\hat{X}|A}(x|a') - p_{X|A}(x|a')). \tag{63}$$

Taking the absolute value on both sides, we have

$$P_a \left| p_{X|A}(x|a) - p_{\hat{X}|A}(x|a) \right| = \left| \sum_{a' \neq a} P_{a'}(p_{\hat{X}|A}(x|a') - p_{X|A}(x|a')) \right| \tag{64}$$

$$\leq \sum_{a' \neq a} P_{a'} \left| p_{\hat{X}|A}(x|a') - p_{X|A}(x|a')) \right|, \tag{65}$$

where Equation (65) follows from the triangle inequality. Thus, it holds that

$$d_{\text{TV}}(p_{X|A}(\cdot|a), p_{\hat{X}|A}(\cdot|a)) = \frac{1}{2} \int \left| p_{X|A}(x|a) - p_{\hat{X}|A}(x|a) \right| dx$$

$$\leq \frac{1}{2} \int \frac{1}{P_a} \sum_{a' \neq a} P_{a'} \left| p_{\hat{X}|A}(x|a') - p_{X|A}(x|a') \right| dx \tag{66}$$

$$= \frac{1}{P_a} \sum_{a' \neq a} P_{a'} \left( \frac{1}{2} \int \left| p_{\hat{X}|A}(x|a') - p_{X|A}(x|a') \right| dx \right) \tag{67}$$

$$= \frac{1}{P_a} \sum_{a' \neq a} P_{a'} d_{\text{TV}}(p_{X|A}(\cdot|a'), p_{\hat{X}|A}(\cdot|a')). \tag{68}$$

This concludes the proof. $\qquad\square$

# G    Face image super-resolution - complementary details and results

## G.1    Synthetic data sets

All the CelebA-HQ images we use are of size $512 \times 512$. The image-to-image translation model we utilize, `stabilityai/stable-diffusion-xl-refiner-1.0`, is sourced from Hugging Face [77] and boasts over 1,200,000 downloads (at the time writing this paper). This model integrates SDXL [67] with SDEdit [53]. For all groups, we adjust the hyperparameters `strength` and `guidance_scale` from their default settings, with `strength` set to 0.4. When translating a CelebA-HQ image $x$ into a group image using its specified text instruction (see Table 2), we choose the *smallest* value from $[8.5, 9.5, 10.5, 11.5, 12.5]$ as the `guidance_scale` hyperparameter, such that the resulting image is classified as belonging to the group. Otherwise, if none of these `guidance_scale` values work for some group (*i.e.*, their class is incorrect), we discard all the translations of $x$ from all groups. To clarify, this means that the translated images for different groups may use different `guidance_scale` values, as long as all translations are correctly classified. The text instructions we use for each group are provided in Table 2. For all groups, we use the same `negative_prompt` text instruction "`ugly, deformed, fake, caricature`". Each of the resulting groups contains 1,356 images of size $512 \times 512$. In Figures 17 to 20 we present 130 image samples from each group.

| Group | Image-to-image translation text instruction |
|---|---|
| Old&Asian | `120 years old human, Asian, natural image, sharp, DSLR` |
| Young&Asian | `20 years old human, Asian, natural image, sharp, DSLR` |
| Old&non-Asian | `120 years old human, natural image, sharp, DSLR` |
| Young&non-Asian | `20 years old human, natural image, sharp, DSLR` |

**Table 2:** Text instructions for the image-to-image translation model to generate images of each fairness group. See Section 4 and Appendix G.1 for more details.

## G.2    Visual results

Visual results of all algorithms (the reconstructions of each fairness group) for $s \in \{4, 8, 16, 32\}$ and $\sigma_N \in \{0, 0.1\}$ are provided in Figures 21 to 28.

## G.3    Additional levels of additive noise

Figure 3 presents quantitative results with all scaling factors, and without adding white Gaussian noise ($\sigma_N = 0$). Here, in Figures 6 and 7 we report the results with $\sigma_N \in \{0.1, 0.25\}$. We observe similar trends and conclusions as in Figure 3 (please refer to Section 4.2 for more details).

## G.4    Comparing GPI$_{FID}$ instead of GPI$_{KID}$

We report in Figures 8 to 10 the GPI$_{FID}$ of each group, where FID is the Fréchet Inception Distance [30]. These results show trends similar to those observed in Figure 3. Namely, using the statistical distance FID instead of KID does not alter the trends and conclusions of the results.

## G.5    Additional group metrics

We report, compare and analyze additional group performance metrics.

**GP$_{NN}$ and GR$_{NN}$**    We approximate the GP and GR of each group using [45], a method which evaluates the precision and recall between two distributions in their feature space. We denote the results by GP$_{NN}$ and GR$_{NN}$, respectively. Note that this approach to approximate GP differs from our previous experiments, where we use the classification hit rate (Figures 3, 6 and 7). Similarly to the experiments where we compute GPI$_{KID}$ (Section 4.2) and GPI$_{FID}$ (Appendix G.4), GP$_{NN}$ and GR$_{NN}$ are computed by extracting image features using the last average pooling layer of the FairFace combined age & ethnicity classifier [35].

**GPSNR and GLPIPS** For each group we compute the Peak Signal-to-Noise Ratio (PSNR) and the Learned Perceptual Image Patch Similarly (LPIPS) [103][3], where these metrics are evaluated by feeding the restoration algorithm only with the group's inputs and with respect to the group's ground truth images. Formally, we define the Group PSNR (GPSNR) and the Group LPIPS (GLPIPS) as

$$\text{GPSNR}(a) = \mathbb{E}[\text{PSNR}(X, \hat{X})|A = a], \tag{69}$$

$$\text{GLPIPS}(a) = \mathbb{E}[\text{LPIPS}(X, \hat{X})|A = a], \tag{70}$$

where the expectation is taken over the joint distribution of a group's ground truth images and their reconstructions, $p_{X,\hat{X}|A}(\cdot, \cdot|a)$.

The results for all noise levels $\sigma_N \in \{0.0, 0.1, 0.25\}$ are provided in Figures 8 to 10. First, note that both the GPSNR and the GLPIPS metrics are unreliable indicators of bias. For example, the metrics GP, $\text{GP}_{\text{NN}}$ $\text{GPI}_{\text{KID}}$, and $\text{GPI}_{\text{FID}}$ all indicate that the group young&non-Asian receives better treatment than the group young&Asian (*e.g.*, the GP of the former group is clearly higher than that of the latter group across all noise levels and scaling factors). However, both groups exhibit roughly similar GPSNR and GLPIPS scores. This highlights why assessing the fairness of image restoration algorithms solely based on GPSNR, GLPIPS or similar metrics (MSE, SSIM [87], *etc.*) might not be sufficient. This result regarding GPSNR is not surprising, as it is well known that such a metric often does not correlate with perceived image quality [10]. Regarding GLPIPS, it might be more effective to use image features extracted by a classifier trained to identify the sensitive attributes in question. We leave exploring this option for future work. Second, the $\text{GP}_{\text{NN}}$ values in Figures 8 to 10 are almost identical to the GP scores reported in Figures 3, 6 and 7. This suggests that approximating the true GP either through the classification hit rate (as in Figures 3, 6 and 7) or via [45] (as done in this section), are consistent. Third, the $\text{GR}_{\text{NN}}$ scores suggest potential unfairness in the perceptual variation across different groups. For example, when $s = 16, \sigma_N = 0$, we observe that all algorithms consistently produce higher $\text{GR}_{\text{NN}}$ scores for the young&non-Asian group compared to the young&Asian group.

## G.6 Feature extractors ablation

We employ the `dinov2-vit-g-14` [62], `clip-vit-l-14` [68], and `inception-v3-compat` [78] feature extractors via `torch-fidelity` [57] to compute the $\text{GPI}_{\text{KID}}$ for each fairness group (previously, we used the image features extracted from the FairFace classifier's final average pooling layer). The results are presented in Figures 11 to 13.

The outcomes from both the `dinov2-vit-g-14` and `clip-vit-l-14` feature extractors generally align with those of the FairFace image classifier, though the biases exposed by these extractors are less pronounced. Put differently, computing $\text{GPI}_{\text{KID}}$ with either of these general-purpose feature extractors leads to a smaller disparity in the $\text{GPI}_{\text{KID}}$ of the different fairness groups. Moreover, the `inception-v3-compat` image feature extractor yields inconsistent results, suggesting that the old&Asian group receives more favorable treatment compared to the old&non-Asian group (contrary to the biases indicated by the other feature extractors). The following section strengthens our argument that this behavior of `inception-v3-compat` is undesirable. Overall, relying on such general-purpose image feature extractors seems unsatisfactory for the purpose of uncovering nuanced biases in face image restoration methods.

## G.7 Considering age and ethnicity as separate sensitive attributes

In Section 4.2 we reveal a significant discrepancy between PF and RDP regarding whether the groups old&Asian and old&non-Asian are treated equally. Specifically, both groups achieve similar GP, while the $\text{GPI}_{\text{KID}}$ of the latter group (old&non-Asian) is notably better (lower) than that of the former group (old&Asian). In other words, $\text{GPI}_{\text{KID}}$ indicates that the old&non-Asian group enjoys a better preservation of ethnicity.

Let us support our claim in Section 4.2 that this outcome of PF is the desired one, by showing that RDP may obscure the fact that some sensitive attributes are treated better than others. Indeed, as shown in Figure 14, the ethnicity of the old&non-Asian group is better preserved than that of the old&Asian group, while Figure 15 confirms that the age of these two groups is equally preserved.

---

[3]Future work may investigate the utility of *no-reference* perceptual quality measures (*e.g.*, [54, 55, 79]) to assess fairness in image restoration.

While RDP fails to uncover this ethnicity bias when the fairness groups are determined based on *both* age and ethnicity, PF clearly reveals it.

### G.8 Final details

All algorithms are evaluated using the official codes and checkpoints provided by their authors. We use the `torch-fidelity` package [57] (GitHub commit `a61422f`) to compute the KID [9], FID [30], precision and recall [45]. The GPSNR and the GLPIPS are computed using the `piq` package [38, 39] (version 0.8.0 in `pip`).

Finally, note that some of the evaluated algorithms generate output images of size $256 \times 256$ (*e.g.*, DDNM), while others produce images of size $512 \times 512$ (*e.g.*, RestoreFormer). Consequently, for fair quantitative evaluations, we resize the outputs of the latter algorithms, along with the ground truth images, to $256 \times 256$. To clarify, the super-resolution scaling factors are calculated based on the $256 \times 256$ image size. For instance, when $s = 4$, the resolution of the input images is $64 \times 64$.

## H  Adversarial attacks - complementary details

The degradation we apply consists of three consecutive steps: (1) Average pooling down-sampling with a scale factor of $s = 4$, (2) additive white Gaussian noise with a standard deviation of $\sigma_N = 0.1$, and then (3) JPEG compression with a quality factor of 50. We attack each degraded image using a tweaked version of the I-FGSM basic attack [14] with $\alpha = 6/255$ and $T = 200$. In particular, instead of using the $L_2$ loss in I-FGSM like in [14], we forward each attacked output through a classifier that predicts the age category of the output face image [56], and then maximize the log-probability of the oldest age group category. In other words, we forward each degraded image through RestoreFormer++ and then feed the result to the age classifier. We then use the I-FGSM update rule to maximize the soft-max probability of the oldest age category (this adversarial attack technique was employed in [60]).

## I  Additional experiments on image denoising and deblurring

We conduct additional experiments on image denoising and deblurring to further demonstrate the utility of the proposed notion of perceptual fairness. Specifically, for image denoising we use additive white Gaussian noise with standard deviation $\sigma_N = 0.5$, and for image deblurring we use a Gaussian blur kernel of size $k = 5$ and $\sigma = 10$, and add to the blurred image a white Gaussian noise of standard deviation $\sigma_N \in \{0.1, 0.25, 0.5\}$. Since these degradations are not handled well by the GAN-based methods, we only compare DPS, DDNM$^+$, DDRM, and PiGDM.

Quantitative results are reported in Figure 16, and visual comparisons are provided in Figures 29 to 32. As in the super-resolution experiments, PF is able to expose bias (which is also visually clear) when RDP fails to do so, and not vice versa.

## J  Computational resources

All our experiments are conducted on a NVIDIA RTX A6000 GPU.

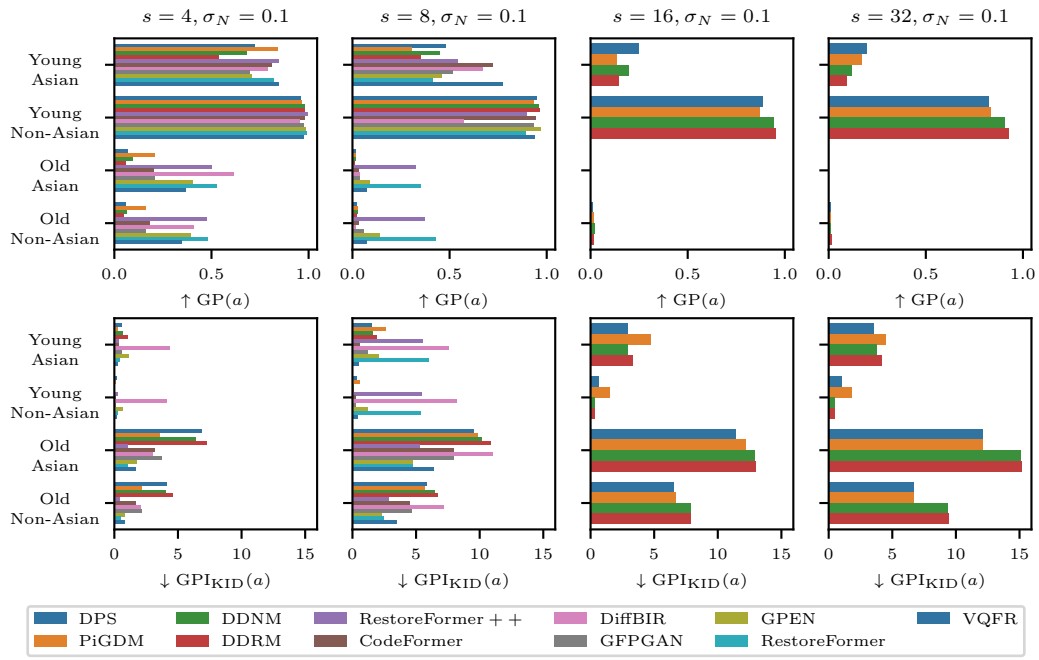

**Figure 6:** Experiments similar to Figure 3, but when the standard deviation of the additive white Gaussian noise is $\sigma_N = 0.1$.

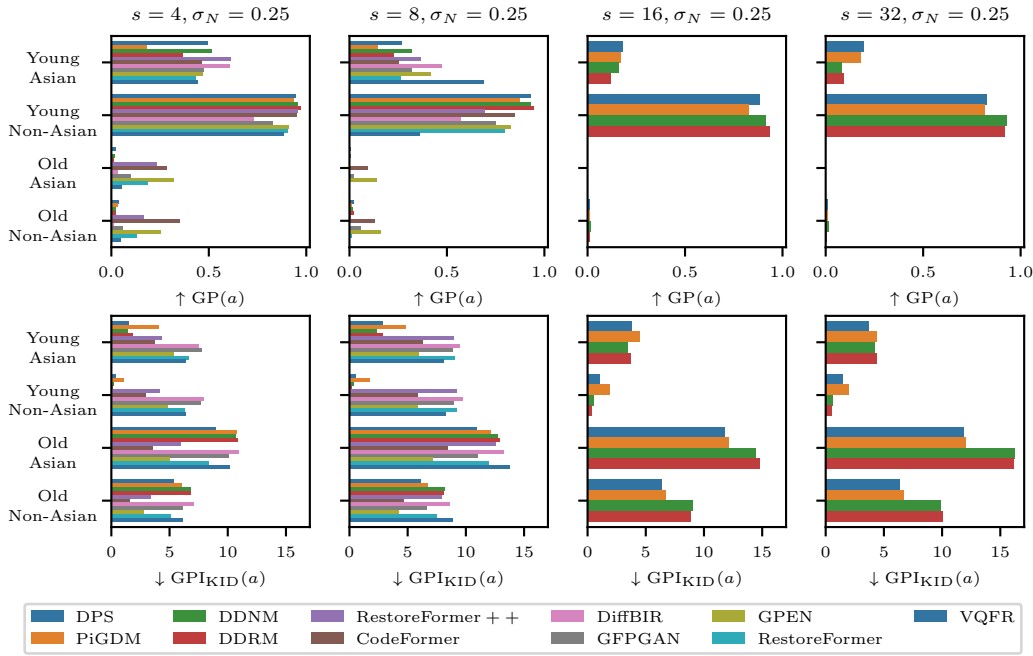

**Figure 7:** Experiments similar to Figure 3, but when the standard deviation of the additive white Gaussian noise is $\sigma_N = 0.25$.

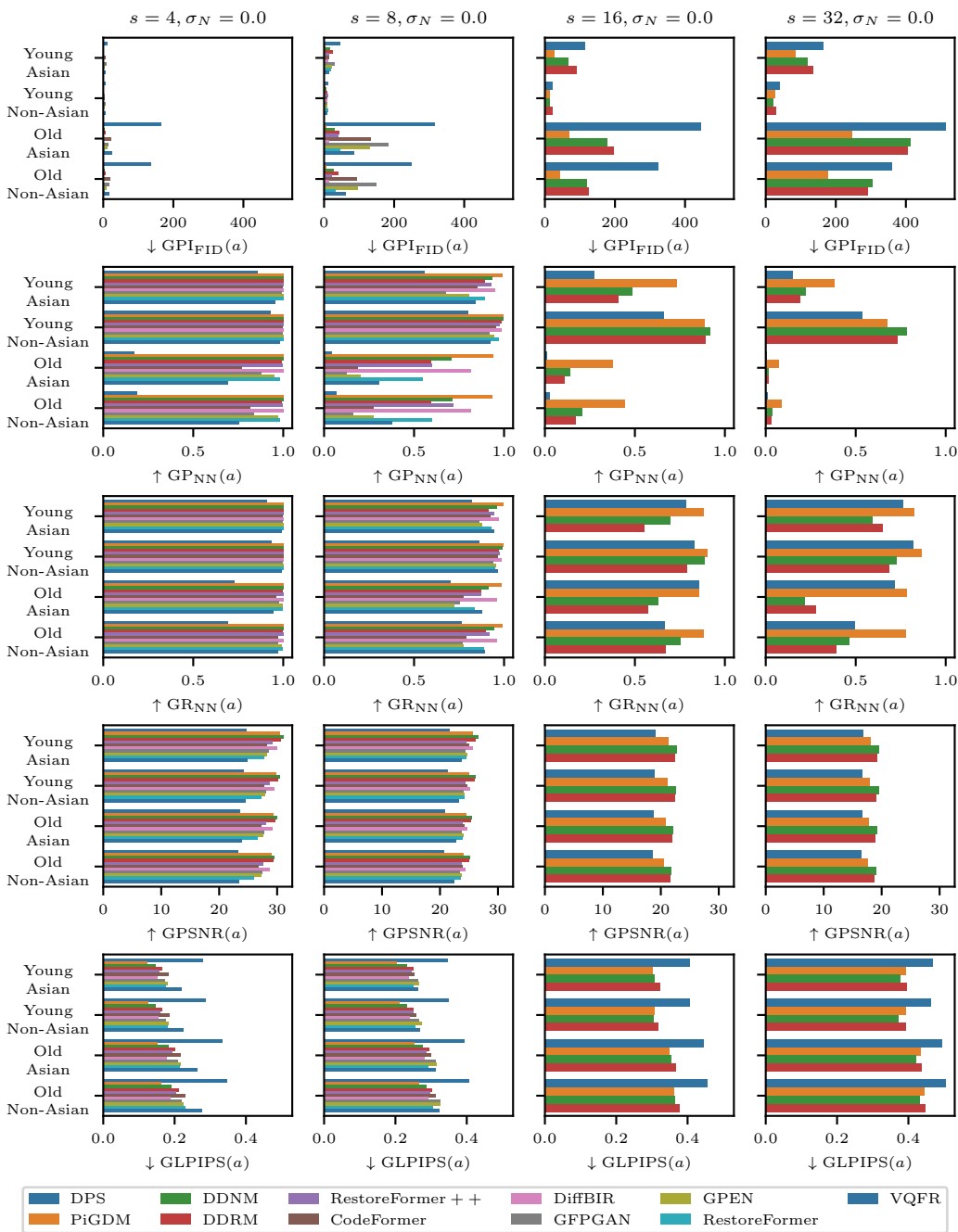

**Figure 8:** Evaluation of additional group metrics where the additive noise level is $\sigma_N = 0.0$ and the super-resolution scaling factor is $s \in \{4, 8, 16, 32\}$. Please refer to Appendix G for more details.

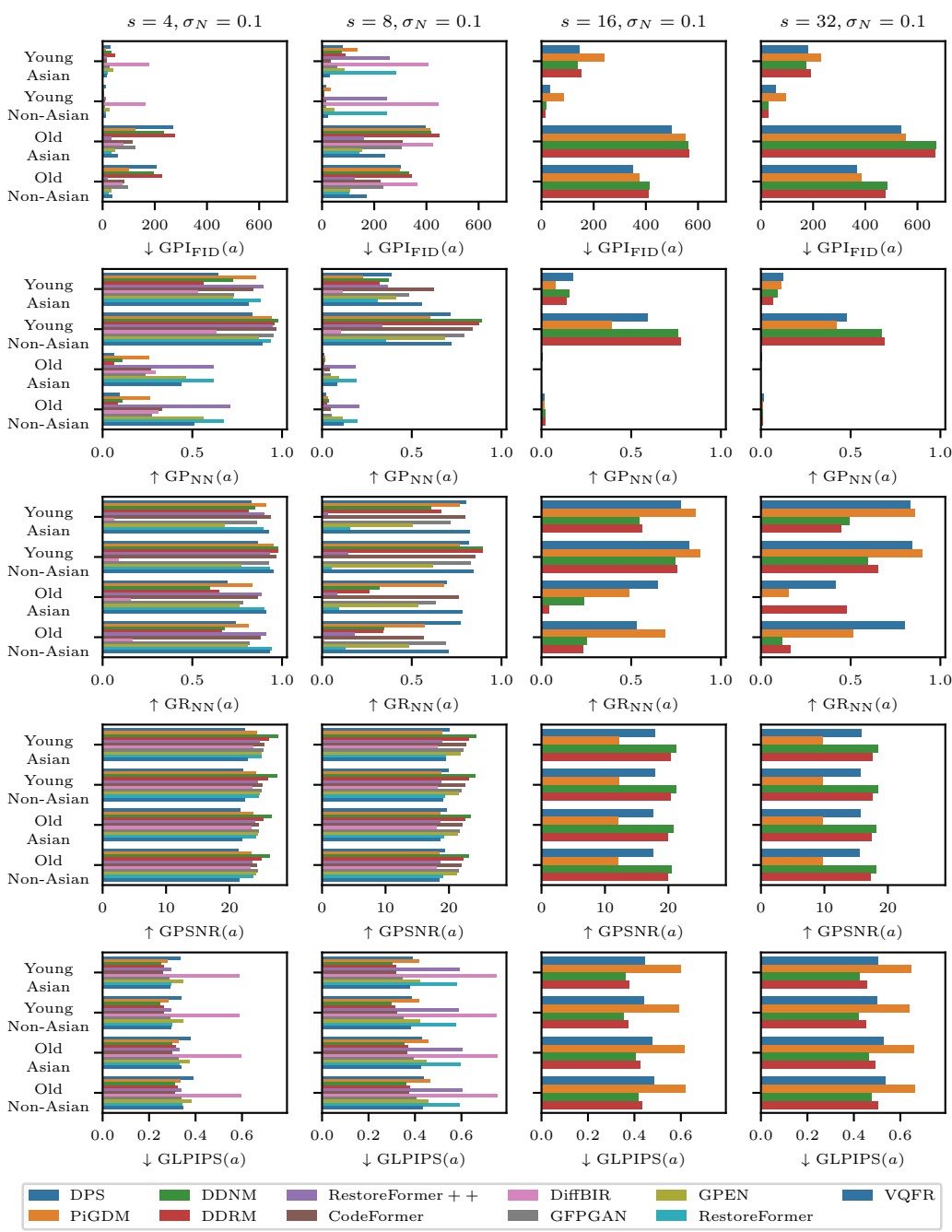

**Figure 9:** Evaluation of additional group metrics where the additive noise level is $\sigma_N = 0.1$ and the super-resolution scaling factor is $s \in \{4, 8, 16, 32\}$. Please refer to Appendix G for more details.

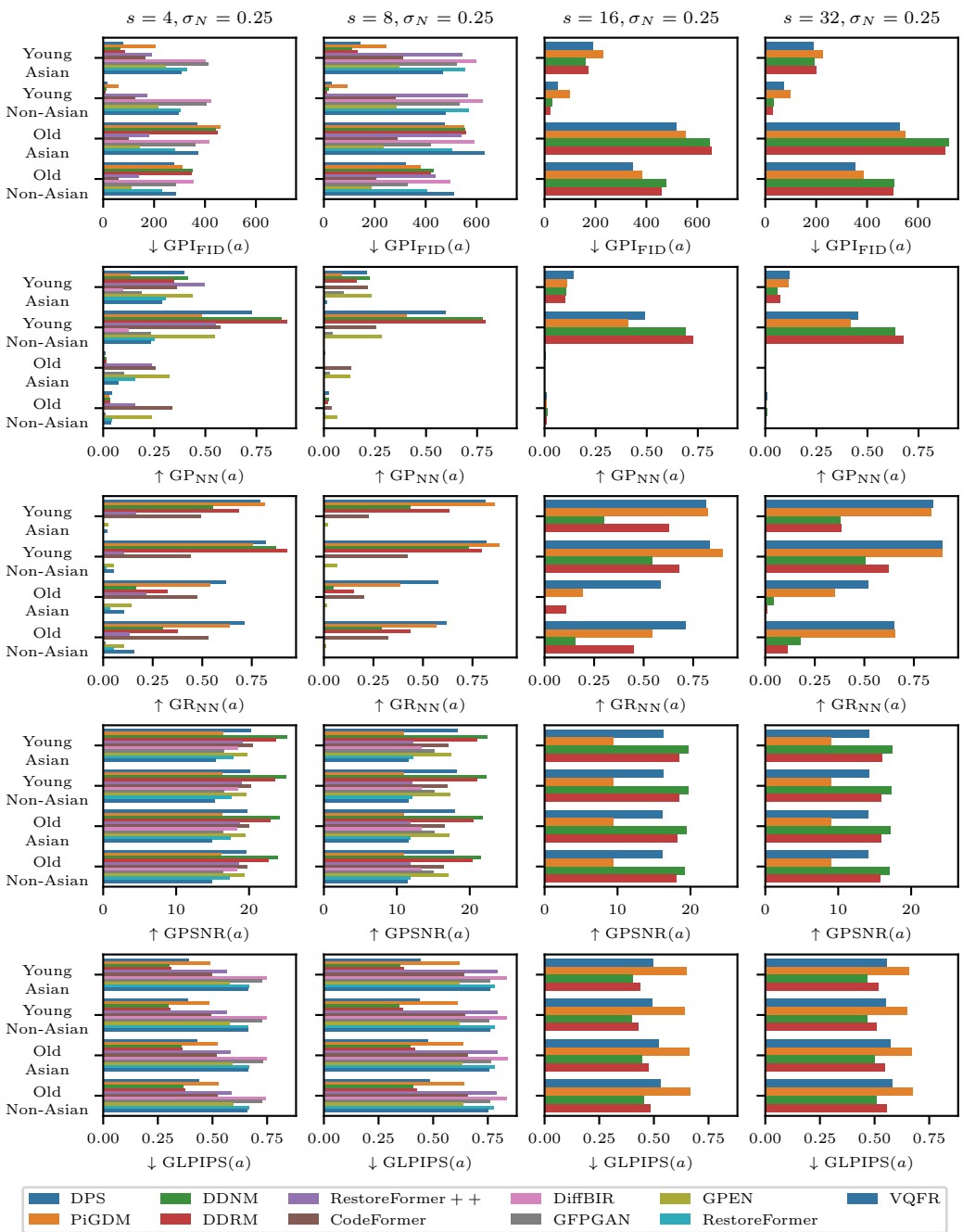

**Figure 10:** Evaluation of additional group metrics where the additive noise level is $\sigma_N = 0.25$ and the super-resolution scaling factor is $s \in \{4, 8, 16, 32\}$. Please refer to Appendix G for more details.

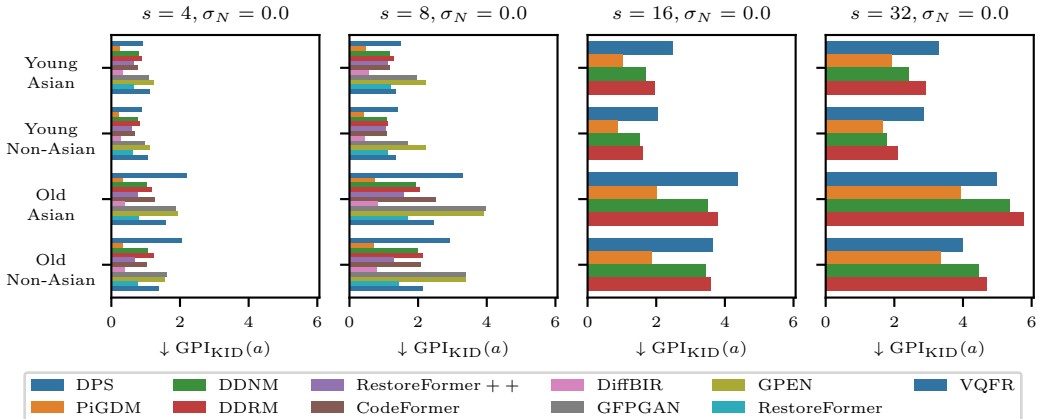

**Figure 11:** Using the `dinov2-vit-g-14` feature extractor [62] via `torch-fidelity` [57] to compute the GPI_{KID} of each group. This general-purpose feature extractor network is somewhat able to detect bias between the old&Asian and old&non-Asian (as detected before by extracting features from the FairFace image classifier). However, the bias is significantly less pronounced in this case.

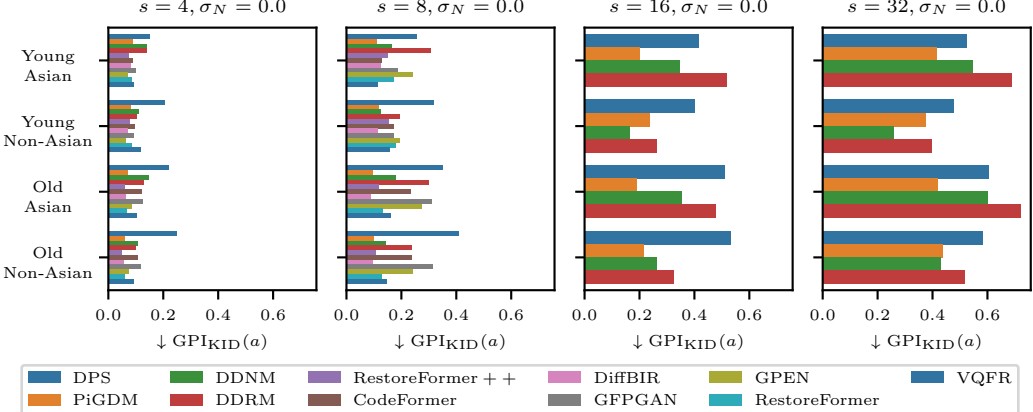

**Figure 12:** Using the `clip-vit-l-14` feature extractor [68] via `torch-fidelity` [57] to compute the GPI_{KID} of each group. Even this general purpose feature extractor network is somewhat able to detect some bias between the old&Asian and old&non-Asian (as detected before by extracting features from the FairFace image classifier). However, the bias is significantly less pronounced in this case.

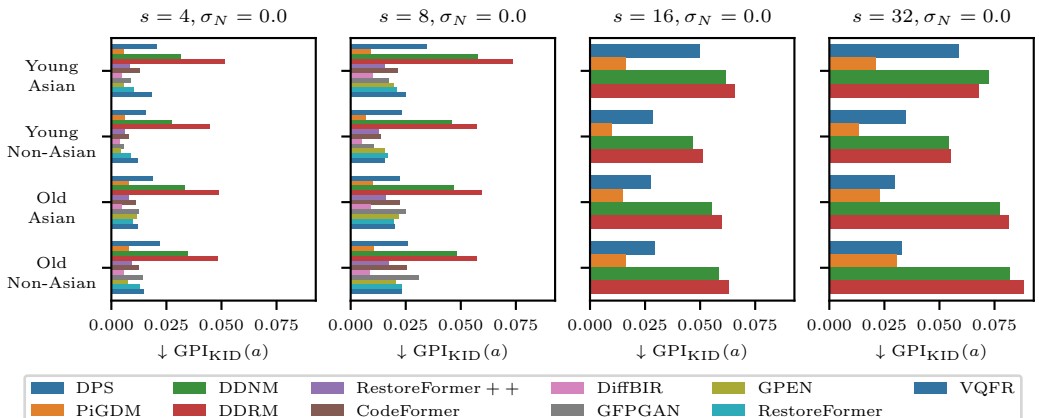

**Figure 13:** Using the `inception-v3-compat` feature extractor [78] via `torch-fidelity` [57] to compute the $\text{GPI}_{\text{KID}}$ of each group. These results of `inception-v3-compat` hint that the old&Asian group in some cases receive *better* treatment than the old&non-Asian group, while all the other feature extractors suggest the opposite bias. This outcome `inception-v3-compat` is also inconsistent with the experiments in Appendix G.7, which demonstrate that the old&non-Asian group is the one receiving the better treatment.

Ethnicity is the only considered sensitive attribute

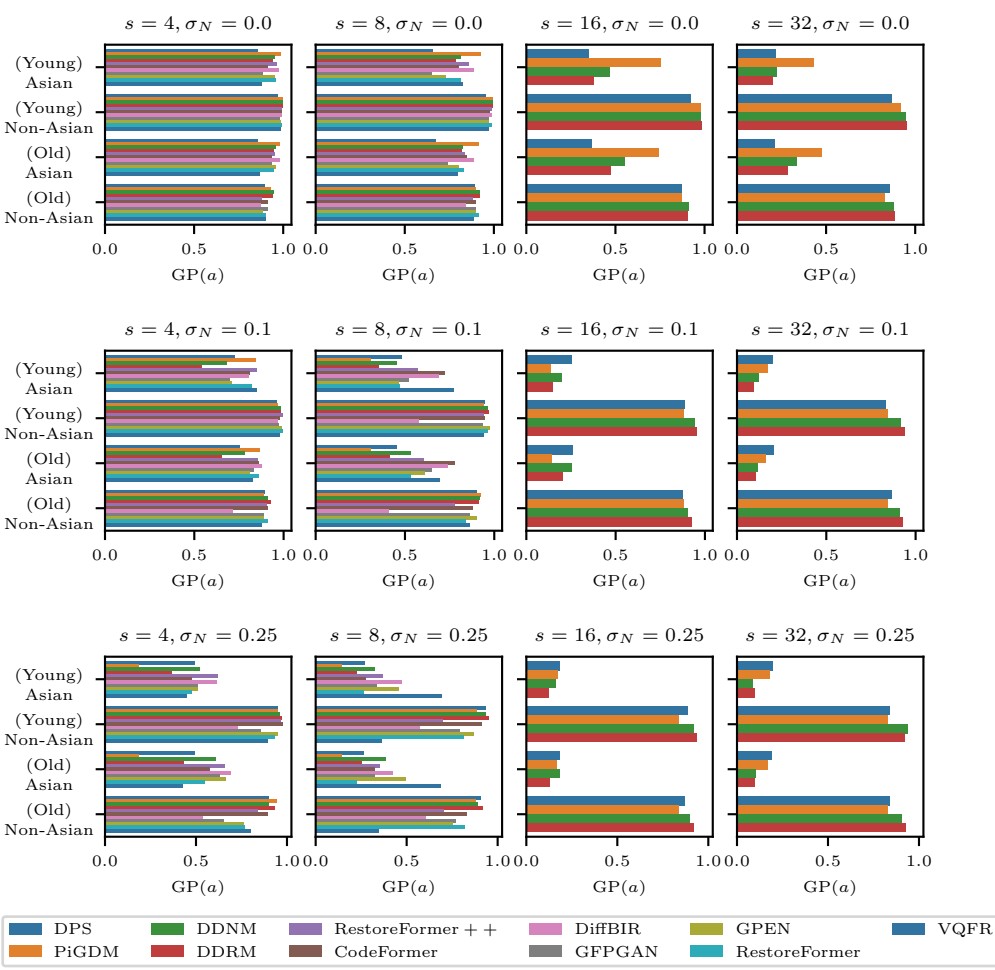

**Figure 14:** Evaluating the GP of each group, where ethnicity is the only considered sensitive attribute. Here, the groups old&Asian and young&Asian are each considered as Asian, and the groups old&non-Asian and young&non-Asian are each considered as non-Asian. For clarity, we still specify in each bar plot the corresponding age of each group, but the classifier operates solely on ethnicity (*i.e.*, the GP is approximated with respect to ethnicity alone). As we claim in Section 4.2, the ethnicity of the old&non-Asian group is clearly preserved better than that of the old&Asian group.

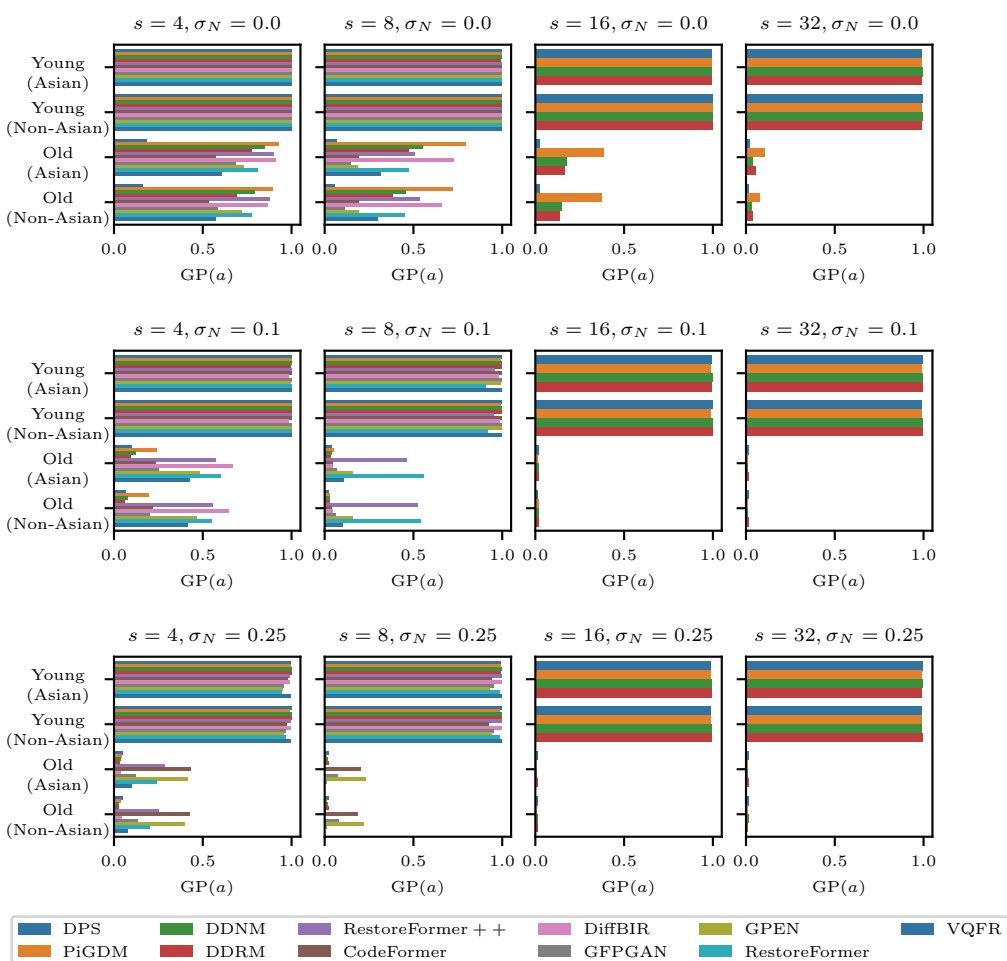

**Figure 15:** Evaluating the GP of each group, where age is the only considered sensitive attribute. Here, the groups old&Asian and old&non-Asian are each considered as old, and the groups young&Asian and young&non-Asian are each considered as young. For clarity, we still specify in each bar plot the corresponding ethnicity of each group, but the classifier operates solely on age (*i.e.*, the GP is approximated with respect to age alone). As we claim in Section 4.2, the age of both the old&non-Asian and old&Asian groups is (roughly) equally preserved.

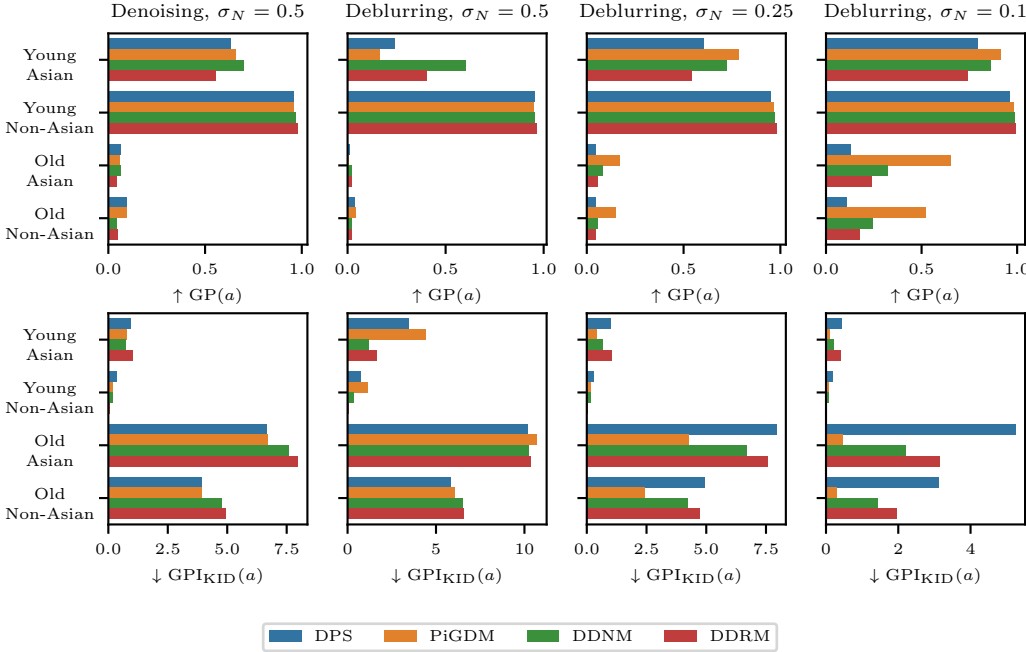

**Figure 16:** Experiments similar to Figure 3, but on the image denoising and deblurring tasks described in Appendix I. We observe similar trends in these tasks as well. Namely, as in the super-resolution tasks, PF exposes a clear bias when RDP does not (but not vice versa).

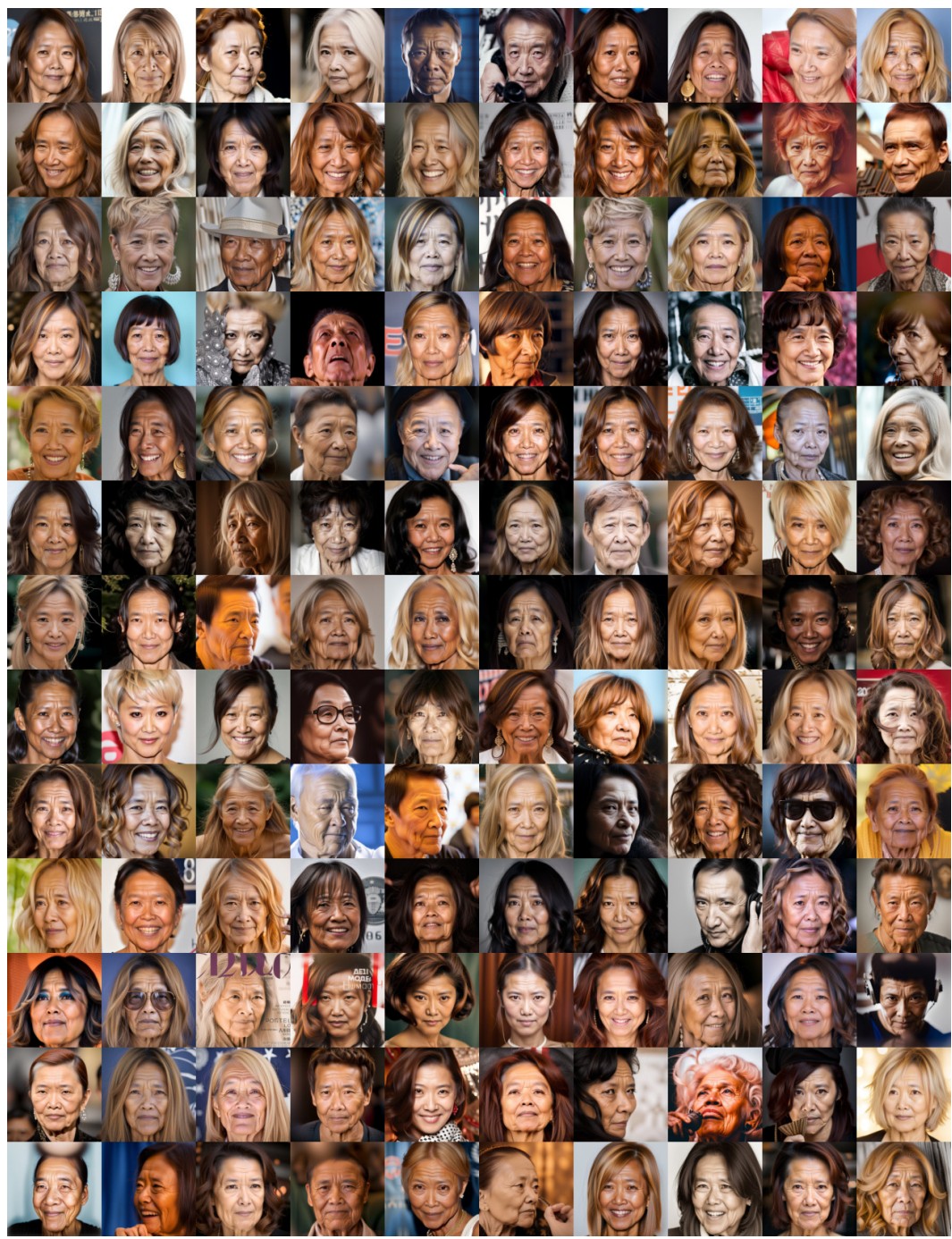

**Figure 17:** Examples of generated images for the old&Asian user group. These samples were generated by passing images from the CelebA-HQ test partition [36] through the SDXL image-to-image model. The text instruction used was ''`120 years old human, Asian, natural image, sharp, DSLR`''. The FairFace ethnicity and age classifier [35] categorizes all of these images as belonging to either the Southeast Asian or East Asian ethnicities, and to the 70+ age group.

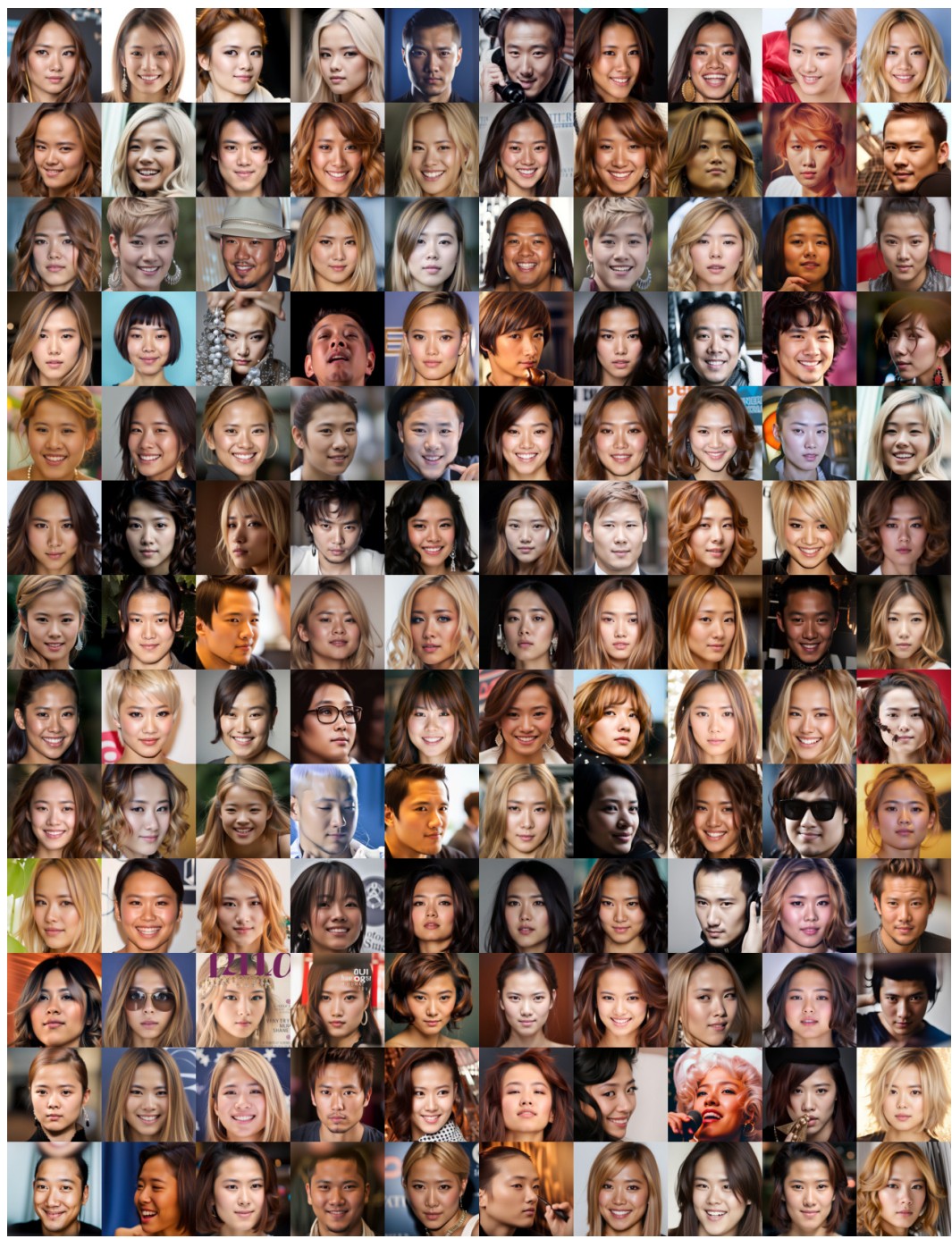

**Figure 18:** Examples of generated images for the young&Asian user group. These samples were generated by passing images from the CelebA-HQ test partition [36] through the SDXL image-to-image model. The text instruction used was ''20 years old human, Asian, natural image, sharp, DSLR''. The FairFace ethnicity and age classifier [35] categorizes all of these images as belonging to either the Southeast Asian or East Asian ethnicities, and to any age group younger than 70 years old.

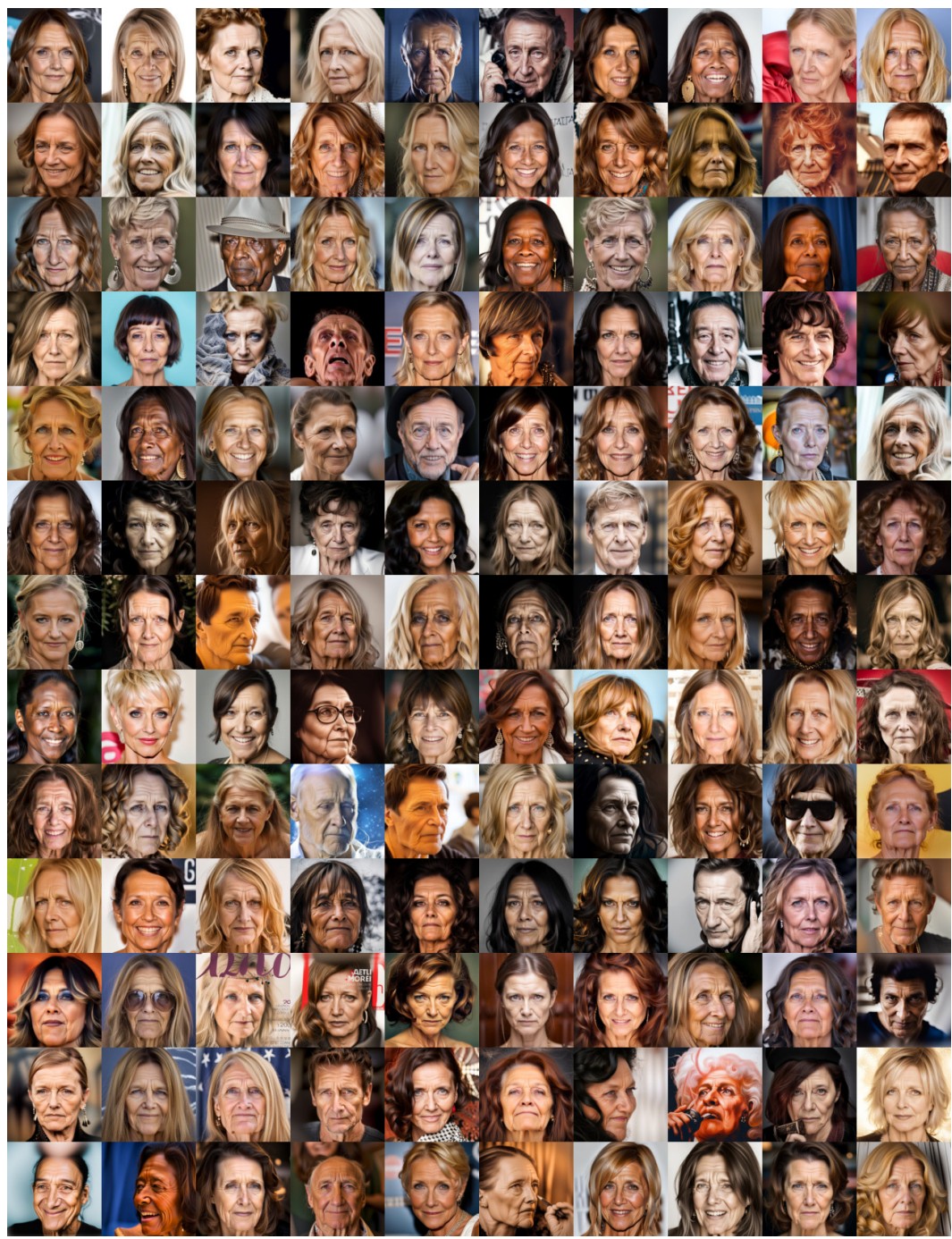

**Figure 19:** Examples of generated images for the old&non-Asian user group. These samples were generated by passing images from the CelebA-HQ test partition [36] through the SDXL image-to-image model. The text instruction used was ''120 years old human, natural image, sharp, DSLR''. The FairFace ethnicity and age classifier [35] categorizes all of these images as belonging to ethnicities other than Southeast Asian or East Asian, and to the 70+ age group.

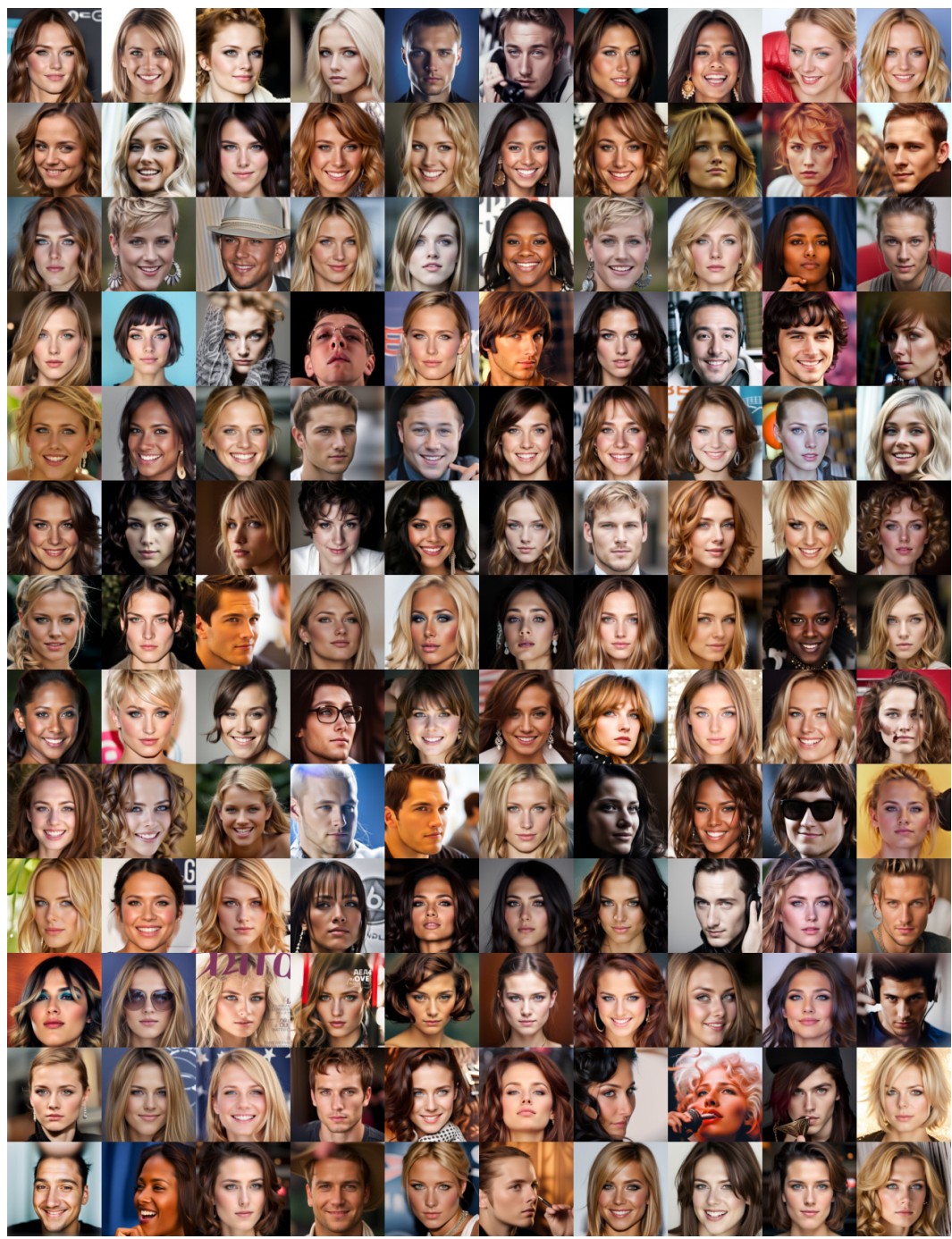

**Figure 20:** Examples of generated images for the young&non-Asian user group. These samples were generated by passing images from the CelebA-HQ test partition [36] through the SDXL image-to-image model. The text instruction used was ''20 years old human, natural image, sharp, DSLR''. The FairFace ethnicity and age classifier [35] categorizes all of these images as belonging to ethnicities other than Southeast Asian or East Asian, and to any age group younger than 70 years old.

Visual results for $s = 4$, $\sigma_N = 0.0$

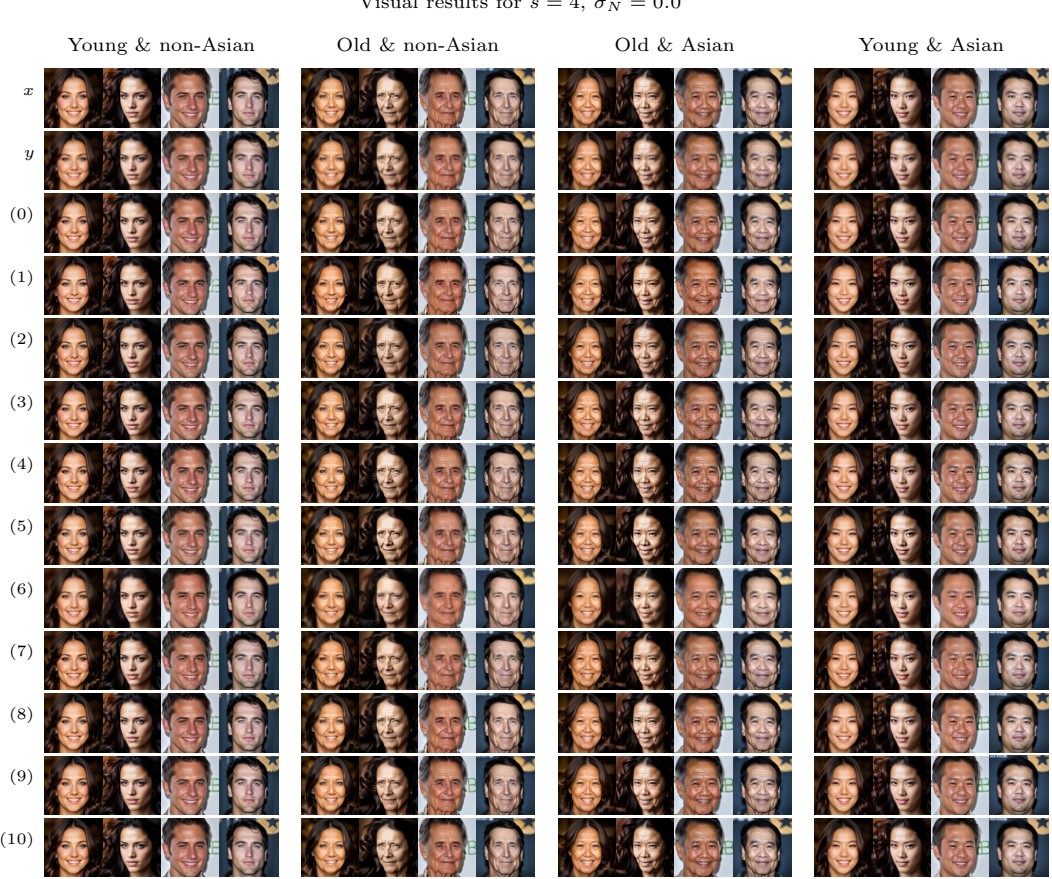

**Figure 21:** Face image super-resolution for each fairness group, where $s = 4, \sigma_N = 0$. (0) DDRM, (1) VQFR, (2) CodeFormer, (3) DDNM$^+$, (4) RestoreFormer $++$, (5) GPEN, (6) DPS, (7) GFPGAN, (8) PiGDM, (9) RestoreFormer, (10) DiffBIR. **Zoom in for best view**.

Visual results for $s = 4$, $\sigma_N = 0.1$

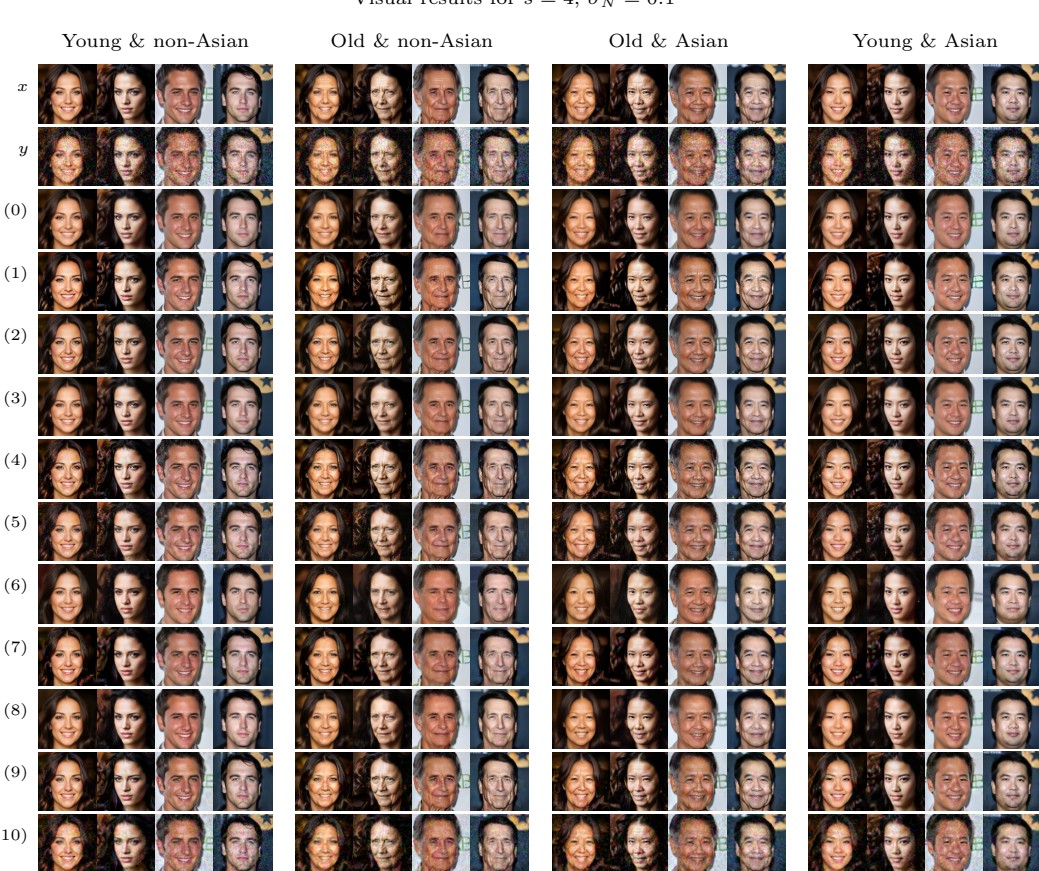

**Figure 22:** Face image super-resolution for each fairness group, where $s = 4, \sigma_N = 0.1$. (0) DDRM, (1) VQFR, (2) CodeFormer, (3) DDNM$^+$, (4) RestoreFormer $++$, (5) GPEN, (6) DPS, (7) GFPGAN, (8) PiGDM, (9) RestoreFormer, (10) DiffBIR. **Zoom in for best view**.

Visual results for $s = 8$, $\sigma_N = 0.0$

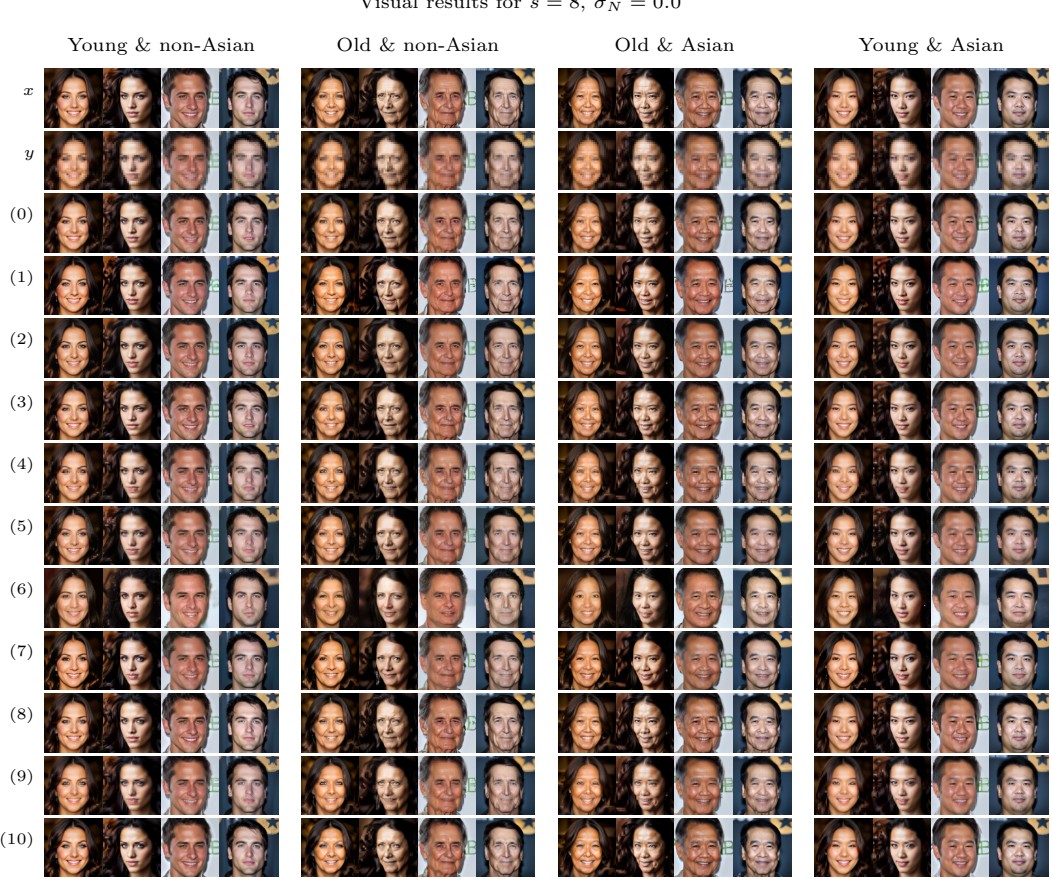

**Figure 23:** Face image super-resolution for each fairness group, where $s = 8, \sigma_N = 0$. (0) DDRM, (1) VQFR, (2) CodeFormer, (3) DDNM$^+$, (4) RestoreFormer $++$, (5) GPEN, (6) DPS, (7) GFPGAN, (8) PiGDM, (9) RestoreFormer, (10) DiffBIR. **Zoom in for best view**.

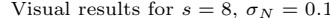

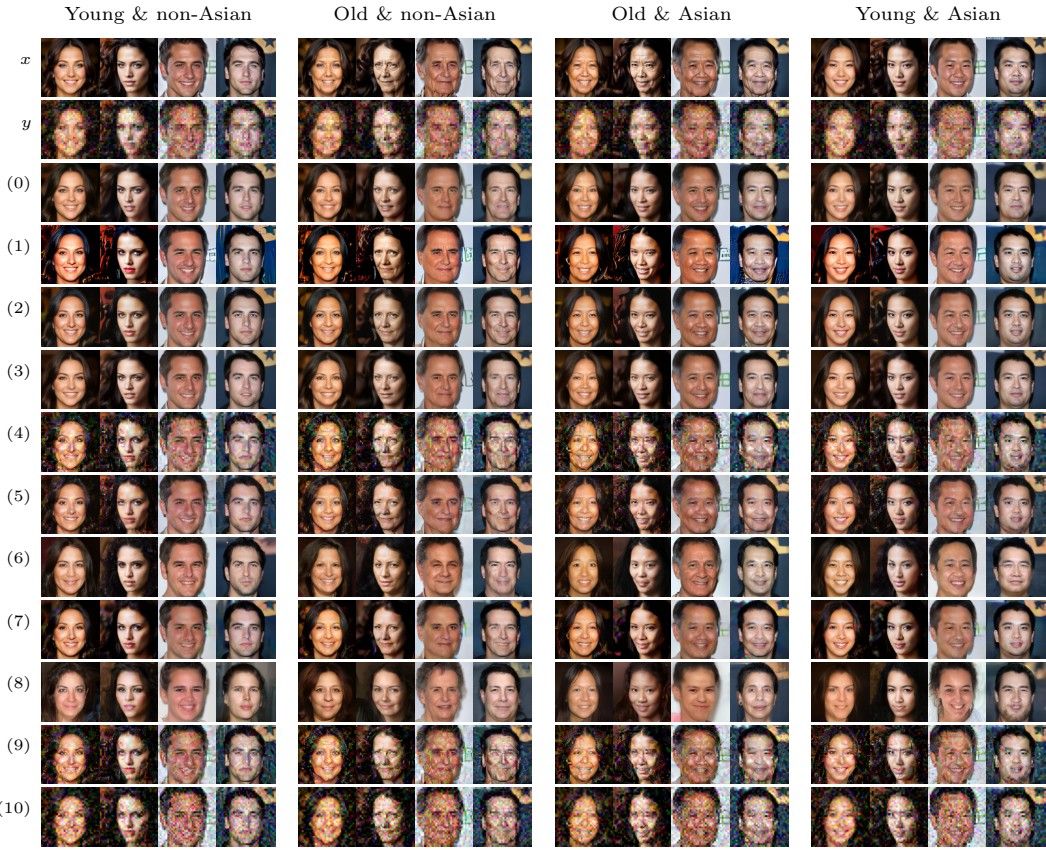

**Figure 24:** Face image super-resolution for each fairness group, where $s = 8, \sigma_N = 0.1$. (0) DDRM, (1) VQFR, (2) CodeFormer, (3) DDNM$^+$, (4) RestoreFormer $++$, (5) GPEN, (6) DPS, (7) GFPGAN, (8) PiGDM, (9) RestoreFormer, (10) DiffBIR. **Zoom in for best view**.

Visual results for $s = 16$, $\sigma_N = 0.0$

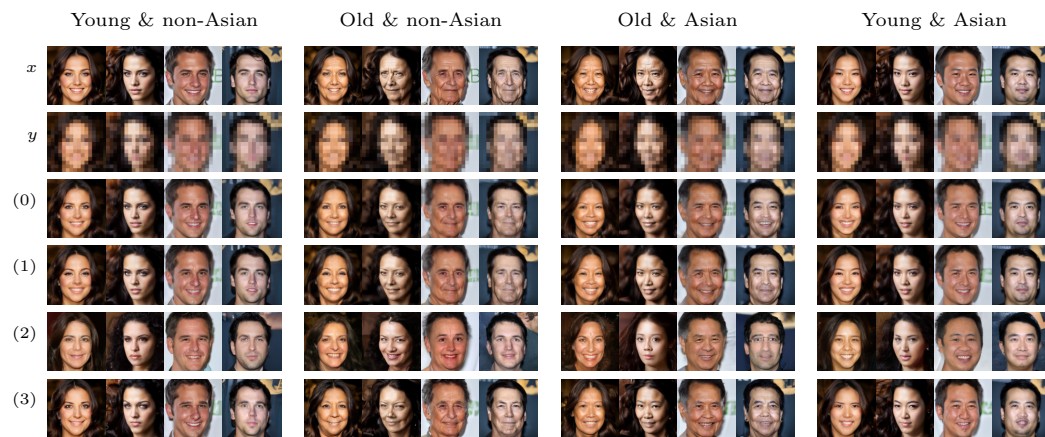

**Figure 25:** Face image super-resolution for each fairness group, where $s = 16, \sigma_N = 0$. (0) DDRM, (1) DDNM$^+$, (2) DPS, (3) PiGDM. **Zoom in for best view**.

Visual results for $s = 16$, $\sigma_N = 0.1$

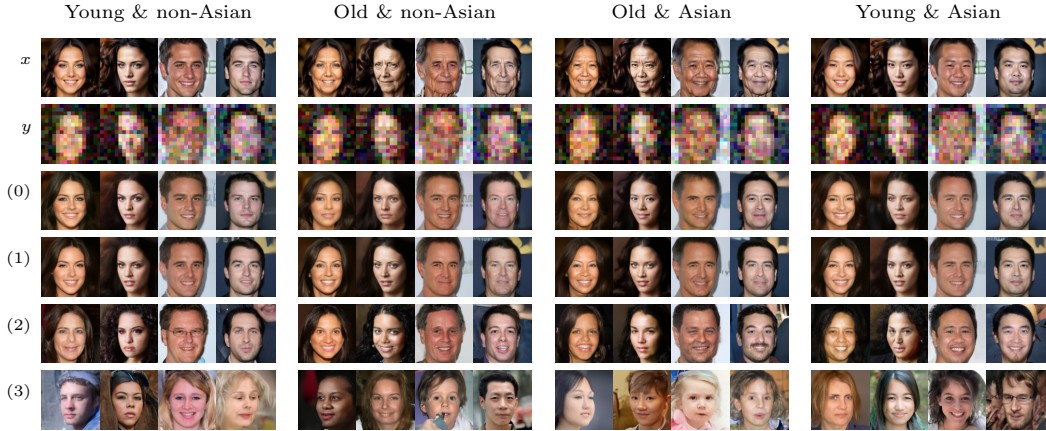

**Figure 26:** Face image super-resolution for each fairness group, where $s = 16, \sigma_N = 0.1$. (0) DDRM, (1) DDNM$^+$, (2) DPS, (3) PiGDM. **Zoom in for best view**.

Visual results for $s = 32$, $\sigma_N = 0.0$

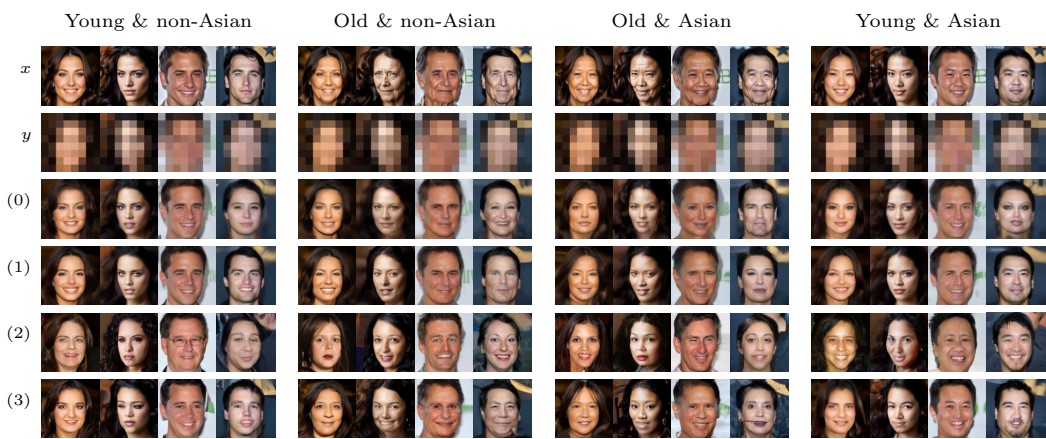

**Figure 27:** Face image super-resolution for each fairness group, where $s = 32, \sigma_N = 0$. (0) DDRM, (1) DDNM$^+$, (2) DPS, (3) PiGDM. **Zoom in for best view**.

Visual results for $s = 32$, $\sigma_N = 0.1$

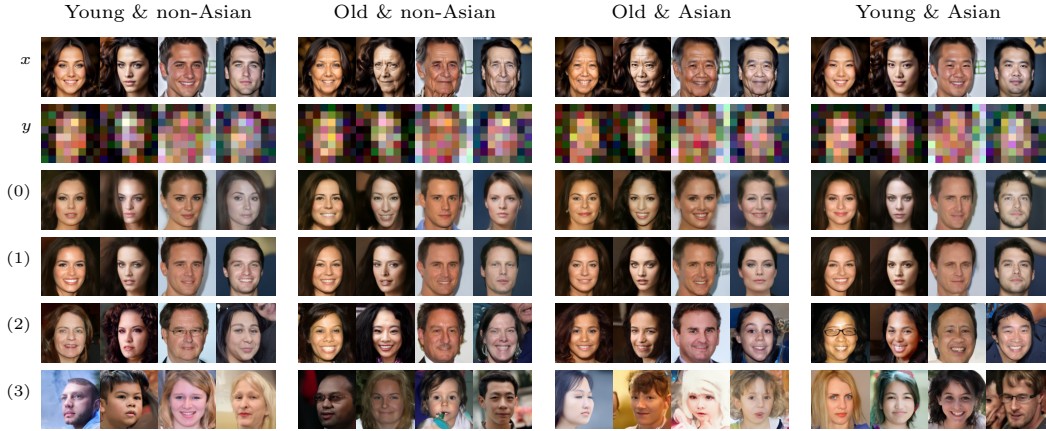

**Figure 28:** Face image super-resolution for each fairness group, where $s = 32, \sigma_N = 0.1$. (0) DDRM, (1) DDNM$^+$, (2) DPS, (3) PiGDM. **Zoom in for best view**.

Visual results for denoising, $\sigma_N = 0.5$

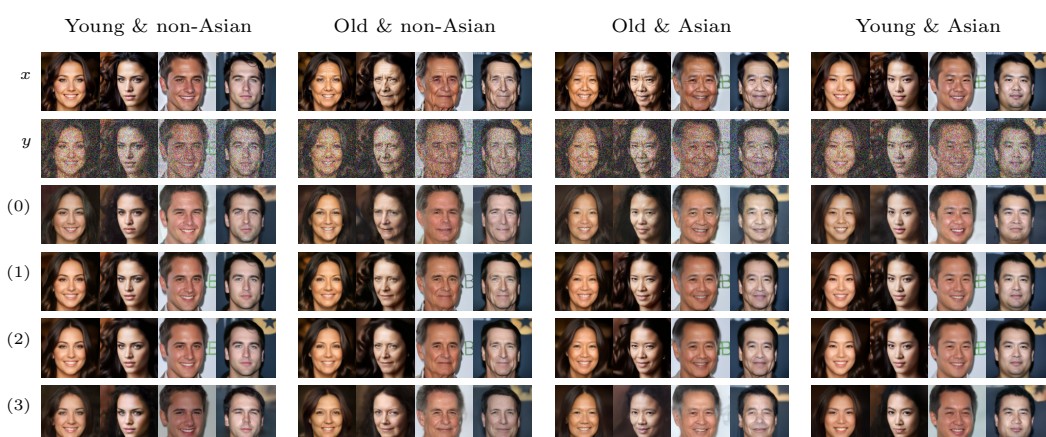

**Figure 29:** Face image denoising for each fairness group, where $\sigma_N = 0.5$. (0) DDRM, (1) DDNM$^+$, (2) DPS, (3) PiGDM. **Zoom in for best view**.

Visual results for deblurring, $\sigma_N = 0.1$

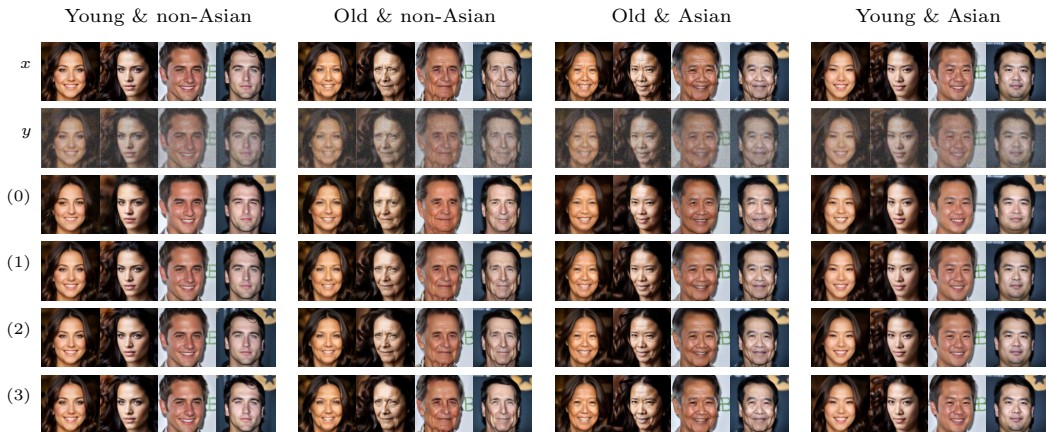

**Figure 30:** Face image deblurring for each fairness group, where $\sigma_N = 0.1$. (0) DDRM, (1) DDNM$^+$, (2) DPS, (3) PiGDM. **Zoom in for best view**.

Visual results for deblurring, $\sigma_N = 0.25$

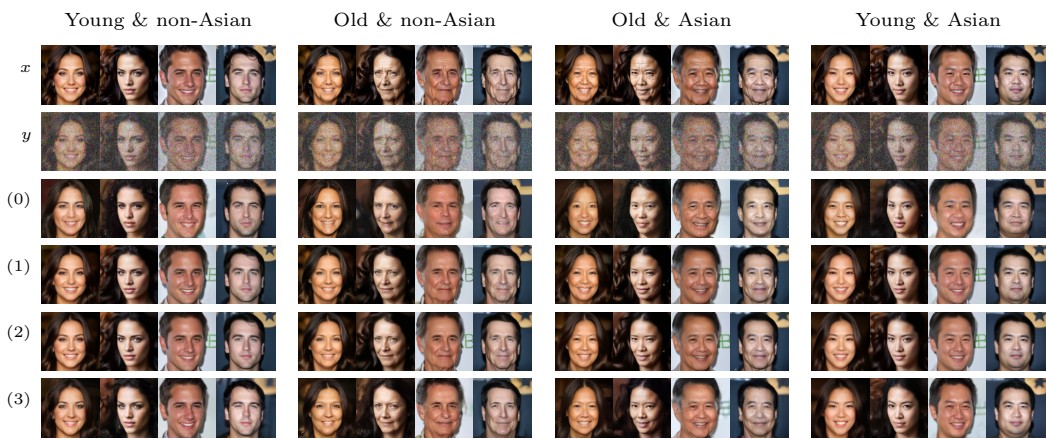

**Figure 31:** Face image deblurring for each fairness group, where $\sigma_N = 0.25$. (0) DDRM, (1) DDNM$^+$, (2) DPS, (3) PiGDM. **Zoom in for best view**.

Visual results for deblurring, $\sigma_N = 0.5$

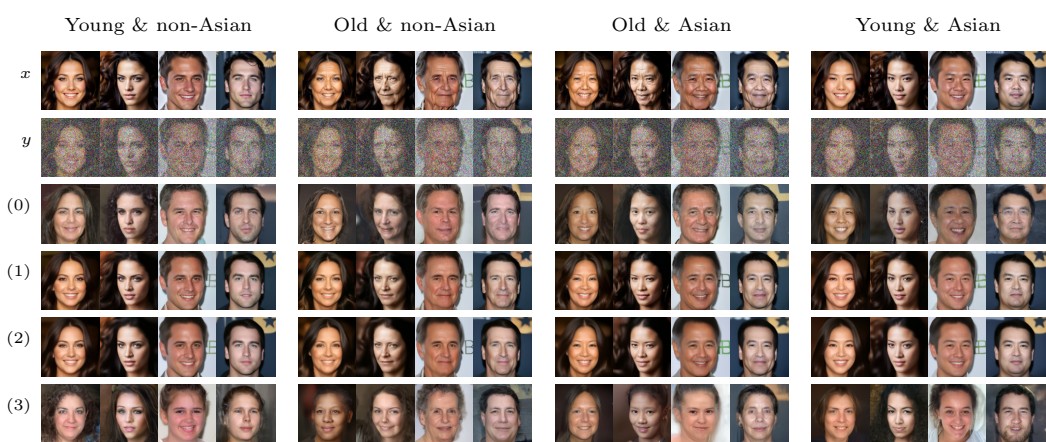

**Figure 32:** Face image deblurring for each fairness group, where $\sigma_N = 0.5$. (0) DDRM, (1) DDNM$^+$, (2) DPS, (3) PiGDM. **Zoom in for best view**.

