# OpenReview forum: "Perceptual Fairness in Image Restoration"
_NeurIPS.cc/2024/Conference — NeurIPS 2024 poster_

### Official Review · Reviewer_ziAr · 2024-07-06

**Soundness:** 3
**Presentation:** 4
**Contribution:** 3
**Rating:** 6
**Confidence:** 3

**Summary:**

This paper considers fairness issue of image restoration and proposes Group Perceptual Index to measure the distance between restoration distribution and gt distribution. Experimental and theoretical results demonstrate that the superiority the proposed perceptual fairness over previous method.

**Strengths:**

1.	The fairness is important for the image restoration community and the topic is interesting to study. The proposed Group Perceptual Index can be reasonable to measure the fairness properly.
2.	The paper includes solid theoretical and experimental results which provides evidence for fairness measurement.
3.	The paper is well-written and easy to follow.

**Weaknesses:**

1. The main results are mainly based face restoration. Can this method be useful for general scenario of image restoration?
2. The paper proposes a measure to detect the fairness issue. But can you suggest some potential solutions to address this problem?

**Questions:**

Can the method be extended to measure other fields, like image generation?

**Limitations:**

Yes, the authors discuss limitations carefully.

---

> ### Author Rebuttal · Authors · 2024-08-02
>
> ###  Can our method be useful for general-content image restoration?
> This is a very interesting point that definitely deserves an explanation. While the proposed GPI is indeed suitable for evaluating fairness in natural images with complex structures, fairness issues are particularly critical when dealing with human images due to their societal implications (which is why we only evaluated facial images). For example, if a general-content image restoration algorithm performs better on images with complex structures than on images of clear skies, this discrepancy is unlikely to be problematic for practitioners, as long as the algorithm attains good performance overall. Moreover, previous works (e.g., [1]) evaluated fairness with respect to non-human subjects (e.g., dogs and cats), but these studies provide limited insights into human-related fairness issues, which often arise due to subtle differences between images. Expanding our method to other datasets remains an avenue for future work. We will include a discussion about this point in the paper.
>
> ### How can we train models to achieve perceptual fairness?
> Please let us propose a method to train algorithms to achieve the best possible perceptual fairness when the sensitive attributes of each ground truth image in the data is known. In particular, one can train a conditional discriminator that takes the sensitive attribute as an additional input, namely, the discriminator is trained to distinguish between the ground truth images of a group (e.g., women images) and their reconstructed images (e.g., the reconstructions produced for degraded images of women). While training such a discriminator, one can add a regularization to the objective of the restoration algorithm (in an "adversarial" fashion) to minimize the deviation in the discriminator scores for different groups. For example, at each training step of the restoration algorithm, one can average the discriminator scores produced for each group and then minimize the standard deviation of the results. This will ensure that the average discriminator scores produced for the reconstructions of each group would be similar for all groups. While this proposed approach was not explored in our paper, it represents an interesting direction for training fair image restoration algorithms. We will thoroughly describe this proposed approach in our manuscript, as a suggestion for future work.
>
> ### Can our method be extended to other fields such as image generation?
> Yes! Thank you for raising this interesting question. The concept of GPI can indeed be extended to other fields, and in particular to image generation. For example, one can regularize a diffusion model (a "time"-dependent image denoiser) during training to attain similar GPIs for all groups. This can be achieved, e.g., by regularizing the denoiser at each time-step using the "adversarial" technique we described above ("*How can we train models to achieve perceptual fairness?*"), while making the discriminator time-dependent as well. Incorporating such a regularization would ensure that the output distribution at each time-step of the diffusion would be balanced with respect to the specified sensitive attributes. This implies that the output distribution at the end of the generation process would also be balanced. We will discuss this interesting potential avenue in our paper.
>
> #### References
>
> [1] Ajil Jalal et al. "Fairness for Image Generation with Uncertain Sensitive Attributes." Proceedings of the International Conference on Machine Learning (ICML), 2021.

---

> > ### Comment · Reviewer_ziAr · 2024-08-09
> > **After Rebuttal**
> >
> > Thanks for the rebuttal. The paper is interesting but also has potential unsolved/undiscussed issues. I insist on my original score.

---

### Official Review · Reviewer_msyK · 2024-07-10

**Soundness:** 2
**Presentation:** 2
**Contribution:** 3
**Rating:** 6
**Confidence:** 1

**Summary:**

This work introduces a new method for assessing fairness in image restoration, called the Group Perceptual Index (GPI). This measure quantifies the statistical difference between the distribution of a group's original images and the distribution of their restored versions. The authors illustrate the effectiveness of GPI by applying it to advanced face image super-resolution algorithms.

**Strengths:**

- the problem tackled in this paper is of practical importance
- paper is written well
- the proposed method is theoretically sound and is shown to work in meaningful ways when used on the problem space of image super-resolution

**Weaknesses:**

- Usefulness of the method is validated only on the super-resolution solution. Given the fact that the proposed method has potential to impact various image restoration algorithms, it would have been interesting to see how well it does on other image restoration application such as image denoising, deblurring etc.
- It is also not clear what kind of changes to the existing super-resolution methods might result in better fairness handling. Some insights into why certain methods are not good at fairness handling as compared to others might have helped the future works.

**Questions:**

Please address my comments under weaknesses

**Limitations:**

Authors has addressed the limitations to a satisfactory extent.

---

> ### Author Rebuttal · Authors · 2024-08-02
>
> ### Demonstration only on image super-resolution tasks
> We opted to illustrate our approach on 12 different super-resolution tasks (4 scale factors and 3 noise levels) simply because of the availability of many methods to compare against on these tasks. Note that this choice aligns with common practice in the field, as related works also focus on image super-resolution [1,2,3], which serves as a standard benchmark due to the availability of numerous algorithms for comparison. Nonetheless, we agree that validating our method on other types of image restoration tasks, such as image denoising or deblurring, or on different modalities like audio, video, or text restoration, would further substantiate its robustness and broad applicability. To address this, we will add experiments with image denoising and deblurring to the appendices and discuss the potential for future work on additional types of degradations and data modalities.
>
> ### What kind of changes to the existing super-resolution methods might result in better fairness handling?
> Please let us propose a method to train algorithms to achieve the best possible perceptual fairness when the sensitive attributes of each ground truth image in the data is known. In particular, one can train a conditional discriminator that takes the sensitive attribute as an additional input, namely, the discriminator is trained to distinguish between the ground truth images of a group (e.g., women images) and their reconstructed images (e.g., the reconstructions produced for degraded images of women). While training such a discriminator, one can add a regularization to the objective of the restoration algorithm (in an "adversarial" fashion) to minimize the deviation in the discriminator scores for different groups. For example, at each training step of the restoration algorithm, one can average the discriminator scores produced for each group and then minimize the standard deviation of the results. This will ensure that the average discriminator scores produced for the reconstructions of each group would be similar for all groups. While this proposed approach was not explored in our paper, it represents an interesting direction for training fair image restoration algorithms. We will thoroughly describe this proposed approach in our manuscript, as a suggestion for future work.
>
> ### Insights into why certain methods are not good at fairness handling compared to others
> Thank you for raising this important point. Please note that our theoretical section (Section 3) attempts to provide such insights. For example, we show that common image restoration algorithms, such as the MMSE point estimate or the posterior sampler, may often lead to poor perceptual fairness. This means that perceptual fairness is not "trivially" acquired by common algorithms. Thus, it is interesting to ask:
> 1. Under which circumstances can some algorithm achieve perfect GPI for all groups simultaneously (Theorem 2)?
> 2. Otherwise, when perfect GPI cannot be achieved for all groups simultaneously, under which circumstances can some algorithm achieve perfect perceptual fairness? (Theorems 3 and 4). For example, from Theorem 4 we learn that, among all the algorithms that achieve perfect Perceptual Index (PI), the better ones in terms of perceptual fairness are those that excel on the "toughest" groups, and this is not directly achieved by methods that simply attain perfect PI (e.g., posterior sampling).
>
> #### References
>
> [1] Ajil Jalal et al., "Fairness for Image Generation with Uncertain Sensitive Attributes." Proceedings of the International Conference on Machine Learning (ICML), 2021.
>
> [2] Yochai Blau and Tomer Michaeli, "The Perception-Distortion Tradeoff." Proceedings of the IEEE/CVF Conference on Computer Vision and Pattern Recognition (CVPR), 2018.
>
> [3] Guy Ohayon et al., "The Perception-Robustness Tradeoff in Deterministic Image Restoration." Proceedings of the International Conference on Machine Learning (ICML), 2024.

---

> > ### Comment · Reviewer_msyK · 2024-08-11
> > **Response to Rebuttal**
> >
> > Thanks for all the clarifications made in the rebuttal. Rebuttal has addressed most of my concerns, hence I will upgrade my original score.

---

### Official Review · Reviewer_HsYQ · 2024-07-10

**Soundness:** 3
**Presentation:** 3
**Contribution:** 3
**Rating:** 6
**Confidence:** 4

**Summary:**

This study presents a novel method to evaluate fairness in image restoration using the Group Perceptual Index (GPI). GPI quantifies the statistical disparity between a group's original images and their restored versions. Fairness is assessed by comparing GPIs across multiple groups, striving for perfect Perceptual Fairness (PF) where all GPI values are identical. The research provides theoretical insights into this innovative fairness concept, drawing comparisons to existing frameworks, and showcases its practical implementation through advanced face image super-resolution algorithms.

**Strengths:**

1. The paper is well-structured and includes sufficient theoretical explanations.
2. The concept of GPI is logically sound.
3. The paper demonstrates that the proposed method outperforms other baseline methods.
4. The paper thoroughly discusses both the advantages and limitations of the proposed method. The advantages highlight the method's effectiveness and potential benefits, while the limitations are clearly outlined, providing a balanced view of its capabilities and areas for improvement.

**Weaknesses:**

1. The authors introduce a novel method to evaluate the fairness of image restoration. However, it is important to note that this method has been validated exclusively on image super-resolution tasks. Further validation on other types of image restoration tasks would be beneficial to demonstrate its broader applicability and robustness.
2. How can sensitive attributes be detected and acquired? The impact of sensitive attributes deserves an in-depth discussion.

**Questions:**

1. What is the primary difference between fairness evaluation and standard image quality assessment metrics like PSNR and SSIM?
2. How does this method contribute to the design of fair and unbiased image restoration algorithms?

**Limitations:**

Limitations have been thoroughly discussed and adequately addressed.

---

> ### Author Rebuttal · Authors · 2024-08-02
>
> ### Demonstration only on image super-resolution tasks
> We opted to illustrate our approach on 12 different super-resolution tasks (4 scale factors and 3 noise levels) simply because of the availability of many methods to compare against on these tasks. Note that this choice aligns with common practice in the field, as related works also focus on image super-resolution [1,2,3], which serves as a standard benchmark due to the availability of numerous algorithms for comparison. Nonetheless, we agree that validating our method on other types of image restoration tasks, such as image denoising or deblurring, or on different modalities like audio, video, or text restoration, would further substantiate its robustness and broad applicability. To address this, we will add experiments with image denoising and deblurring to the appendices and discuss the potential for future work on additional types of degradations and data modalities.
>
> ### How can sensitive attributes be detected and acquired?
> Thank you for raising this interesting question. Please note that our work assumes that the sensitive attributes are prespecified, which is the case handled by most works that tackle fairness concerns. It should be noted that the term "sensitive" here is application dependent. For example, in one application gender may be considered the only sensitive attribute, whereas in another application age may be considered sensitive as well. This is despite the fact that both applications use the precise same data for training. Thus, sensitive attributes cannot be detected automatically from the data. Rather, they should be specified by the user according to the particular societal concerns in question. We will discuss this interesting point in the manuscript.
>
> ### The difference between GPI and standard metrics like PSNR
> We appreciate this question and address it in Appendix G.5 (L794 onwards), where we show that such metrics are not good indicators of perceptual bias. Indeed, our experiments (Figures 8-10 in the appendix) show that, while the different groups attain roughly the same distortion (e.g., PSNR, LPIPS), their GPIs differ significantly, a discrepancy that is also visually evident.
>
> ### How can our method contribute to the design of fair and unbiased image restoration algorithms?
> Please let us propose a method to train algorithms to achieve the best possible perceptual fairness when the sensitive attributes of each ground truth image in the data is known. In particular, one can train a conditional discriminator that takes the sensitive attribute as an additional input, namely, the discriminator is trained to distinguish between the ground truth images of a group (e.g., women images) and their reconstructed images (e.g., the reconstructions produced for degraded images of women). While training such a discriminator, one can add a regularization to the objective of the restoration algorithm (in an "adversarial" fashion) to minimize the deviation in the discriminator scores for different groups. For example, at each training step of the restoration algorithm, one can average the discriminator scores produced for each group and then minimize the standard deviation of the results. This will ensure that the average discriminator scores produced for the reconstructions of each group would be similar for all groups. While this proposed approach was not explored in our paper, it represents an interesting direction for training fair image restoration algorithms. We will thoroughly describe this proposed approach in our manuscript, as a suggestion for future work.
>
> #### References
>
> [1] Ajil Jalal et al., "Fairness for Image Generation with Uncertain Sensitive Attributes." Proceedings of the International Conference on Machine Learning (ICML), 2021.
>
> [2] Yochai Blau and Tomer Michaeli, "The Perception-Distortion Tradeoff." Proceedings of the IEEE/CVF Conference on Computer Vision and Pattern Recognition (CVPR), 2018.
>
> [3] Guy Ohayon et al., "The Perception-Robustness Tradeoff in Deterministic Image Restoration." Proceedings of the International Conference on Machine Learning (ICML), 2024.

---

> > ### Comment · Reviewer_HsYQ · 2024-08-11
> >
> > Considering the feedback from other reviewers and the authors' responses, I retain the original rating of weak accept.

---

### Official Review · Reviewer_xBjq · 2024-07-11

**Soundness:** 3
**Presentation:** 4
**Contribution:** 4
**Rating:** 6
**Confidence:** 4

**Summary:**

This paper reveals that the conventional definition of fairness for image restoration is restrictive and often causes controversy. To address this issue, the authors introduce a new approach to measure fairness in image restoration tasks by proposing the Group Perceptual Index (GPI). Specifically, they propose assessing the fairness of an algorithm by comparing the GPI of different groups, where perfect Perceptual Fairness (PF) is achieved if the GPIs of all groups are identical. They theoretically study this notion of fairness and demonstrate its utility on state-of-the-art face image super-resolution algorithms.

**Strengths:**

1. The paper reveals the existing fairness measures such as Representation Demographic Parity (RDP) and highlights their limitations. It shows that these measures can be overly simplistic and may not detect subtle biases that affect different groups.
2. The paper proposes the Group Perceptual Index (GPI) as a measure of fairness in image restoration, which is a novel and significant contribution.
3. It provides a theoretical analysis of the properties of GPI and its relationship to other fairness measures.
4. The authors use a variety of datasets and experimental setups to demonstrate the effectiveness of GPI, which are convincing.

**Weaknesses:**

1. Group Perceptual Index (GPI) also increases the complexity of the evaluation process of image restoration algorithms, compared with the traditional fairness method, because it involves comparing the distributions of different groups.
2. The experiments use synthetic datasets generated from high-quality, aligned face image datasets like CelebA-HQ.
3. The paper only evaluate the proposed method on the face dataset and does not provide the results on other kinds of image data.

**Questions:**

1. The paper shows that achieving perfect GPI for all groups is often not feasible, especially under severe degradation conditions.  it leaves an open question of how to balance fairness among different groups in practice best when perfect GPI cannot be achieved.
2. In addition to face synthesis, is the proposed GPI suitable for natural images with complex structures?
3. The application scenes are not clear. How do we apply the proposed GPI to real-world applications?

**Limitations:**

The paper rethinks the fairness in image restoration and proposes a novel method, called Group Perceptual Index (GPI), to measure the fairness for image restoration models. The proposed method can effectively detect subtle and malicious biases enhances the robustness and security of image restoration systems

---

> ### Author Rebuttal · Authors · 2024-08-02
>
> ### Complexity of evaluating perceptual fairness
> Thank you for raising this important point. We acknowledge that computing the GPI of each group increases the complexity of evaluating fairness compared to previous methods, which typically compute the classification hit rates (e.g., counting the reconstructed images classified as having a specified sensitive attribute). However, it is important to note that both our method and previous methods require each sample to be processed through a classifier. In our method, the classifier is used to extract deep image features to compute metrics such as FID and KID, whereas in previous methods the classifier evaluates the predicted sensitive attributes of each image.
> Thus, the additional computational overhead of our method arises from two factors:
> 1. Extracting deep features from the source images in addition to the reconstructed images, effectively doubling the computation required for feature extraction compared to previous methods.
> 2. Approximating a statistical divergence (e.g., FID) between the extracted features of the ground truth images and the extracted features of the reconstructed images. This additional step introduces some computational overhead (e.g., computing the empirical mean and covariance matrix of the samples, as in FID), but it is relatively minor in practice compared to the benefits of our more nuanced fairness evaluation. We will point this limitation in our paper.
>
> ### Why do we use synthetic datasets?
> We appreciate the reviewer's concern regarding our use of synthetic data sets. Please note that, as discussed in Section 4.1 of our paper, we opted to use synthetic, high-quality datasets because existing face datasets often lack ground truth labels for sensitive attributes such as ethnicity, age, and gender. Moreover, the datasets that do include such labels are typically imbalanced w.r.t. these attributes. To adequately approximate and compare the GPI of different groups, it is essential to have equal amounts of data from each group. Otherwise, approximating the GPI with metrics like FID would lead to completely different scoring scales for varying sample sizes (see, e.g., Figure 1 in [1], which shows that FID suffers from bias when using small sample sizes). Generating synthetic datasets allows us to control the number of images from each group, and therefore allows to ensure balanced and fair evaluations.
>
> ### Applicability to other types of data besides facial images
> This is a very interesting point that definitely deserves an explanation. While the proposed GPI is indeed suitable for evaluating fairness in natural images with complex structures, fairness issues are particularly critical when dealing with human images due to their societal implications (which is why we only evaluated facial images). For example, if a general-content image restoration algorithm performs better on images with complex structures than on images of clear skies, this discrepancy is unlikely to be problematic for practitioners, as long as the algorithm attains good performance overall. Moreover, previous works (e.g., [2]) evaluated fairness with respect to non-human subjects (e.g., dogs and cats), but these studies provide limited insights into human-related fairness issues, which often arise due to subtle differences between images. Expanding our method to other datasets remains an avenue for future work. We will include a discussion about this point in the paper.
>
> ### How to balance fairness among different groups in practice?
> Please let us propose a method to train algorithms to achieve the best possible perceptual fairness when the sensitive attributes of each ground truth image in the data is known. In particular, one can train a conditional discriminator that takes the sensitive attribute as an additional input, namely, the discriminator is trained to distinguish between the ground truth images of a group (e.g., women images) and their reconstructed images (e.g., the reconstructions produced for degraded images of women). While training such a discriminator, one can add a regularization to the objective of the restoration algorithm (in an "adversarial" fashion) to minimize the deviation in the discriminator scores for different groups. For example, at each training step of the restoration algorithm, one can average the discriminator scores produced for each group and then minimize the standard deviation of the results. This will ensure that the average discriminator scores produced for the reconstructions of each group would be similar for all groups. While this proposed approach was not explored in our paper, it represents an interesting direction for training fair image restoration algorithms. We will thoroughly describe this proposed approach in our manuscript, as a suggestion for future work.
>
> ### How do we apply the proposed GPI to real-world applications?
> Many practical imaging systems should ensure that their algorithms do not introduce or amplify biases against any particular group. Mobile phones, for example, are used by everyone, and all of them incorporate image restoration algorithms within the Image Signal Processing (ISP) pipeline to produce a high-quality image from the given sensor measurements. When the degradation is sufficiently severe (e.g., in low-light conditions), even a well-performing image restoration algorithm (e.g., posterior sampler) may treat some groups better than others. Thus, it is important to evaluate the fairness of such algorithms. For example, the GPI can help identify subtle biases that traditional methods might overlook, thereby alerting for fairness issues in these systems.
>
> #### References
> [1] Mikołaj Bińkowski et al. "Demystifying MMD GANs." Proceedings of the International Conference on Learning Representations (ICLR), 2018.
>
> [2] Ajil Jalal et al., "Fairness for Image Generation with Uncertain Sensitive Attributes." Proceedings of the International Conference on Machine Learning (ICML), 2021.

---

> > ### Comment · Reviewer_xBjq · 2024-08-11
> >
> > Thank you for your responses and comprehensive explanation. Your responses partially resolve my concerns, so I keep my original score unchanged. Overall, I like this paper and appreciate the efforts of the authors on the fairness of image restoration, which is inspiring.

---

### Author Rebuttal · Authors · 2024-08-02

# Thank you!
We are deeply grateful to all the reviewers for dedicating their time to evaluate our paper. The feedback we received has been highly encouraging and has helped us improve the quality of our work.

---

### Decision · Program_Chairs · 2024-09-25

**Decision:**

Accept (poster)

**Comment:**

Overall, the reviewers agree that the paper makes a novel and significant contribution by proposing the Group Perceptual Index (GPI) as a measure of fairness in image restoration. The reviewers commend the convincing experiments, logically sound concept, theoretical analysis of GPI's properties, thorough discussion of both the advantages and limitations, and the overall quality of the presentation.

The main concerns are whether the proposed metric is applicable to general images beyond human faces and the sole validation on the image super-resolution task. The rebuttal addresses the first issue and promises to include additional experiments on image denoising and deblurring in the appendices. It also commits to discussing the potential for future work on additional types of degradations and data modalities to address the latter concern. Additionally, to address the ethics issue, the authors agree to highlight the choice of sensitive attributes in the study. Please take these commitments into account in your revision.